# Contextual Combinatorial Bandits With Changing Action Sets Via Gaussian Processes

**Andi Nika**[†]                                                                     *andinika@mpi-sws.org*
*Max Planck Institute for Software Systems*
*Germany*

**Sepehr Elahi**[†]                                                                   *sepehr.elahi@epfl.ch*
*Department of Computer and Communication Sciences*
*EPFL*
*Switzerland*

**Cem Tekin**                                                                 *cemtekin@ee.bilkent.edu.tr*
*Department of Electrical and Electronics Engineering*
*Bilkent University*
*Turkey*

**Reviewed on OpenReview:** *https://openreview.net/forum?id=2RgfAY3jnI&*

## Abstract

We consider a contextual bandit problem with a combinatorial action set and time-varying base arm availability. At the beginning of each round, the agent observes the set of available base arms and their contexts and then selects an action that is a feasible subset of the set of available base arms to maximize its cumulative reward in the long run. We assume that the mean outcomes of base arms are samples from a Gaussian Process (GP) indexed by the context set $\mathcal{X}$, and the expected reward is Lipschitz continuous in expected base arm outcomes. For this setup, we propose an algorithm called Optimistic Combinatorial Learning and Optimization with Kernel Upper Confidence Bounds (O'CLOK-UCB) and prove that it incurs $\tilde{O}(\sqrt{\lambda^*(K)KT\gamma_{KT}(\cup_{t \leq T}\mathcal{X}_t)})$ regret with high probability, where $\gamma_{KT}(\cup_{t \leq T}\mathcal{X}_t)$ is the maximum information gain associated with the sets of base arm contexts $\mathcal{X}_t$ that appeared in the first $T$ rounds, $K$ is the maximum cardinality of any feasible action over all rounds, and $\lambda^*(K)$ is the maximum eigenvalue of all covariance matrices of selected actions up to time $T$, which is a function of $K$. To dramatically speed up the algorithm, we also propose a variant of O'CLOK-UCB that uses sparse GPs. Finally, we experimentally show that both algorithms exploit inter-base arm outcome correlation and vastly outperform the previous state-of-the-art UCB-based algorithms in realistic setups.

## 1 Introduction

The multi-armed bandit (MAB) problem is a cornerstone of reinforcement learning, capturing the fundamental challenge of sequential decision-making under uncertainty: an agent repeatedly chooses actions and refines its beliefs to maximize cumulative reward (Auer et al., 2002b; Bubeck & Cesa-Bianchi, 2012; Auer et al., 2002a). Among its many extensions, two have been especially influential due to their broad practical relevance—from recommender systems to personalized medicine—namely contextual bandits and combinatorial semi-bandits. In contextual bandits, the agent observes side information (context) at the beginning of each round and uses it to guide its decision-making (Lu et al., 2010; Langford & Zhang, 2007; Slivkins,

---

[†] Work done while authors were at Bilkent University.

2011; Chu et al., 2011). In combinatorial semi-bandits, the agent selects a subset of base arms and receives feedback on each selected arm, thereby enabling richer action spaces and outcome observations (Cesa-Bianchi & Lugosi, 2012; Chen et al., 2013; 2016a; Kveton et al., 2015c;a; Zong et al., 2016; Hiranandani et al., 2020). Importantly, in many applications, the set of available arms itself may vary over time—so-called changing action sets (Chakrabarti et al., 2008; Li et al., 2019; Bnaya et al., 2013; Kleinberg et al., 2010).

This paper focuses on contextual combinatorial multi-armed bandits with changing action sets (C3-MAB), a framework that naturally captures repeated interactions in dynamic environments. In each round $t$, the agent observes the available base arms and their contexts, selects a subset (a super arm), receives a reward, and observes noisy outcomes for the chosen arms. We allow base arm availability to change arbitrarily across rounds, and thus analyze regret under any sequence of availability patterns. At the same time, for any given base arm and context, outcomes are drawn from a fixed distribution parameterized by that context. The central challenge is to maximize cumulative reward without knowledge of future context arrivals or the underlying outcome function, requiring the agent to adaptively balance exploration and exploitation in real time.

When the set of possible contexts is large, learning without additional structure becomes infeasible. To overcome this, we impose smoothness via Gaussian processes (GPs), which offer a flexible yet powerful modeling assumption that enables generalization across contexts (Contal et al., 2013). Specifically, we assume the base arm outcome function is a sample from a GP with a known kernel, ensuring both tractability and expressiveness.

GP-based bandit algorithms have a long history of addressing the exploration–exploitation dilemma, beginning with the seminal work of Srinivas et al. (2012). Their closed-form posterior mean and variance yield high-probability confidence bounds on expected rewards, enabling principled action selection. A key advantage of GP-UCB over classical UCB methods lies in its ability to assign meaningful uncertainty estimates even to unseen actions, thanks to posterior smoothness. This property is particularly valuable in environments with time-varying action sets, as it naturally encodes the structure required for efficient learning.

## 1.1 Our contributions

Below we describe the main contributions of this paper.

We propose a new learning algorithm called *Optimistic Combinatorial Learning and Optimization with Kernel Upper Confidence Bounds* (O'CLOK-UCB), tailored to solve the C3-MAB problem over compact context spaces under the assumption that the expected base arm outcomes are samples from a GP and the expected reward is Lipschitz for the expected base arm outcomes. O'CLOK-UCB incurs $\tilde{O}\left(\sqrt{\lambda^*(K)KT\gamma_{KT}(\cup_{t\leq T}\mathcal{X}_t)}\right)$ regret in $T$ rounds in the general case when the feasible action cardinalities are bounded variables, where $\gamma_{KT}(\cup_{t\leq T}\mathcal{X}_t)$ is the maximum information gain with respect to the sequence of $T$ available context sets $\mathcal{X}_t$, to be explicitly defined later, over the feasible actions with cardinalities upper bounded by $K \in \mathbb{N}$, and $\lambda^*(K)$ is the maximum eigenvalue of all covariance matrices of the selected actions up to time $T$.

Our work resolves new theoretical and technical difficulties introduced as a result of simultaneously taking into consideration several bandit features.

- The semi-bandit feedback necessitates a batch update of all the base arm indices at once after all base arms are selected. This, in return, requires a new way of upper-bounding the expected regret, substantially different from previous related work (Srinivas et al., 2012). In order to account for a non-identity covariance matrix (due to dependencies among base arms in an action), we leverage the conditional entropy of Gaussians and properties of eigenvalues (see Lemma 1). Therefore, our setting is not a straightforward extension of a GP bandit in $KT$ rounds.

- We adapt the classical notion of maximum information gain to the C3-MAB setting (represented by $\gamma_{KT}(\cup_{t\leq T}\mathcal{X}_t)$), which takes into account time-varying base arm availability, and provide new regret upper bounds based on this notion of maximum information gain.

- We perform semi-synthetic crowdsourcing simulations on a real-world dataset and compare our algorithm with the previous state-of-the-art. Moreover, we illustrate how the time complexity of O'CLOK-UCB can be improved by using the sparse approximation method, where we only use a subset of the past history of selected contexts to update the posterior. We experimentally show that our method outperforms the previous state-of-the-art UCB-based algorithms in this setting.

## 1.2 Related work

The Combinatorial MAB has been widely studied and well understood in the last decade (Cesa-Bianchi & Lugosi, 2012; Chen et al., 2013; Combes et al., 2015; Kveton et al., 2015c; 2014; Chen et al., 2014), together with its various branches such as cascading bandits (Kveton et al., 2015b), and combinatorial bandits with probabilistically triggered arms (Chen et al., 2016b; Huyuk & Tekin, 2019).

In recent years, the Contextual Combinatorial Multi-armed Bandit (CCMAB) problem has been attracting a lot of interest (Chen et al., 2018; Qin et al., 2014). Chen et al. (2018) studies the CCMAB problem with cascading feedback; that is, the learning agent can observe base arm outcomes of only a prefix of the played super arm. Their model allows for position discounts and a wide range of reward functions. Under monotonicity and Lipschitz continuity of the expected reward, their algorithm, C$^3$-UCB, incurs $\tilde{O}(\sqrt{KT})$ regret in $T$ rounds, where $K$ is the maximum cardinality of any feasible super arm.

The contextual combinatorial multi-armed bandits with changing action sets has been studied in (Chen et al., 2018; Nika et al., 2020). Chen et al. (2018) assumes the expected reward to be submodular, and the expected base arm outcomes are assumed to be Hölder continuous with respect to their corresponding contexts. Their proposed algorithm (CCMAB) addresses the changing arm sets over rounds by uniformly discretizing the context space into a predetermined number of hypercubes (which depends on the time horizon $T$), thus utilizing the outcome similarities between nearby contexts. CCMAB incurs $\tilde{O}(T^{(2\eta+D)/(3\eta+D)})$ regret where $\eta$ is the Hölder constant and $D$ the context space dimension.

A potential drawback of this approach is the fixed discretization of the context space, which implies limited exploitation of the arms' similarity information. Nika et al. (2020) addresses this issue by adaptively discretizing the $D$-dimensional context space $(\mathcal{X}, \|\cdot\|_2)$ following a tree structure, under some mild structural assumptions on $(\mathcal{X}, \|\cdot\|_2)$ (where $\|\cdot\|_2$ is the Euclidean norm) and Lipschitz continuity assumption both of the expected reward with respect to the expected base arm outcomes and the latter with respect to their associated contexts. Their algorithm (ACC-UCB) incurs $\tilde{O}(T^{(\bar{D}+1)/(\bar{D}+2)+\epsilon})$ regret for any $\epsilon > 0$, where $\bar{D}$ is the approximate optimality dimension of the context space $\mathcal{X}$, tailored to capture both the benignness of the base arm arrivals and the structure of the expected reward under a combinatorial setup. In general, $\bar{D} \leq D$, implies an improvement of the bound rates to the previous work.

Adaptive discretization has been utilized in the GP bandit setting by Shekhar & Javidi (2018) for black-box function optimization. Under the same structural assumptions of the arm space, $(\mathcal{X}, d)$ as in (Nika et al., 2020) (where $d$ is a general metric associated with $\mathcal{X}$), and under some Hölder continuity assumptions on their covariance function, they propose a tree-based algorithm whose aim is to maximize the cumulative reward given a fixed budget of samples. They provide both near-optimality dimension type and information type regret bounds. It is worth mentioning that when the context space is very large, and the arm set is time-varying, adaptive discretization has proved to be an efficient technique, yielding optimal regret bounds (Bubeck et al., 2011). Nevertheless, if we further assume that the number of arms that come in a round is finite, imposing a GP prior on the base arm outcomes removes both the need for adaptive discretization and Lipschitz continuity of the expected base arm outcomes. The posterior's smoothness encodes enough information to estimate with high probability the outcomes of any available base arm, regardless of its history of arrivals. In light of this, we approach the C3-MAB problem using a simple procedure under mild assumptions.

The main difference between adaptive discretization, and our GP-based method is that, while adaptive discretization discretizes the search space into regions and maintains statistics over each region, GPs offer a functional approach. Such an approach allows us to retrieve point-wise statistical information in a fine-grained way just by evoking the posterior mean and variance. On the other hand, adaptive discretization operates on a lower resolution, maintaining the same statistical information for all points in the same region.

Table 1: *Comparison with related works.* Below, $K$ represents the fixed cardinality of a feasible super arm; $\lambda^*(K)$ denotes the maximum eigenvalue of all covariance matrices of selected actions up to time $T$; $\overline{D}$ is the approximate optimality dimension introduced in (Nika et al., 2020) and their bounds hold for any $\epsilon > 0$; $\eta$ is the Hölder constant and $\nu$ is the Matérn parameter. The bounds are shown up to polylog factors.

| Work | Context space | Function | Smoothness assumption | Oracle (approx.) | Contextual & Changing action sets | Regret bounds |
|---|---|---|---|---|---|---|
| (Chen et al., 2013) | Finite | Lipschitz | Explicit | $(\alpha, \beta)$ | No | $\tilde{O}\left(\sqrt{KT}\right)$ |
| (Chen et al., 2018) | Infinite | Submodular | Explicit | $(1-1/e)$ | Yes | $\tilde{O}\left(KT^{(D+2\eta)/(D+3\eta)}\right)$ |
| (Nika et al., 2020) | Compact | Lipschitz | Explicit | $\alpha$ | Yes | $\tilde{O}\left(KT^{(\overline{D}+1)/(\overline{D}+2)+\epsilon}\right)$ |
| **Ours** (Linear kernel) | Compact | Lipschitz | GP-induced | $\alpha$ | Yes | $\tilde{O}\left(\sqrt{\lambda^*(K)KDT}\right)$ |
| **Ours** (RBF kernel) | Compact | Lipschitz | GP-induced | $\alpha$ | Yes | $\tilde{O}\left(\sqrt{\lambda^*(K)KDT\log^D T}\right)$ |
| **Ours** (Matérn kernel) | Compact | Lipschitz | GP-induced | $\alpha$ | Yes | $\tilde{O}\left(\lambda^*(K)T^{(D+\nu)/(D+2\nu)}\right)$ |

Consequently, adaptive discretization uses less computational resources, since it focuses only on 'relevant' regions and adaptively refines them based on historical information, while sacrificing resolution. In contrast, GPs offer a high-resolution picture of all relevant statistics over the search space, thus allowing for a more precise inference, with an additional computational cost. We have resolved this latter issue by proposing a practical version of our algorithm. Furthermore, we have provided an experimental comparison between GPs and adaptive discretization, thus showcasing the benefit of using GPs as opposed to the latter.

More recently, Elahi et al. (2023) extended the C3-MAB setting by incorporating group constraints. In their work, base arms belong to groups, and the selected super arm must satisfy constraints based on rewards associated with these groups. They utilize a multi-output Gaussian Process to model distinct outcomes for the primary super arm reward and the group constraint evaluation. Their proposed TCGP-UCB algorithm uses a double-UCB approach to explicitly balance reward maximization and group constraint satisfaction. While tackling a different objective involving constraint satisfaction, their work also leverages GPs for C3-MAB and arrives at a regret bound form similar to ours.

Other works which are closest to ours include (Sandberg et al., 2025). The problem they consider, namely the combinatorial volatile Gaussian process semi-bandit problem, is the same as ours. In addition to regret bounds which have similar dependence, we also adapt the classical notion of information gain to the C3MAB setting. Moreover, while they consider the notion of expected (Bayesian cumulative) regret, we provide high-probability upper bounds on the notion of $\alpha$-approximation regret. Additionally, (Accabi et al., 2018) and (Nuara et al., 2022) propose algorithms with a similar rationale to O'CLOK-UCB. However, while our algorithm is provably no-regret in the C3-MAB setting with (potentially) infinitely many context-dependent time-varying arms, the setting considered in (Accabi et al., 2018) and (Nuara et al., 2022) is a standard combinatorial semi-bandit setting with a context-free time-invariant finite arm set.

In Table 1, we compare characteristics and regret bounds between our work and previous work when the cardinality of feasible super arms is constant. Note that we achieve a substantial improvement from (Nika et al., 2020) and (Chen et al., 2018) on the order $T$ in the case of linear and squared exponential kernels.

The rest of the paper is organized as follows. In Section 2 we lay down our setup and recall some definitions of Gaussian processes and their properties; in Section 3 we introduce our proposed algorithm and explain how it works; in Section 4 we provide the regret bounds for O'CLOK-UCB; Section 5 contains the experimental results, and we conclude in Section 6.

## 2 Problem formulation

We first introduce the general notation to be used throughout the paper. Let us fix a positive integer $K \geq 1$ and a $D$-dimensional vector space $\mathcal{X}$. We write $[K] = \{1, 2, \ldots, K\}$. Vectors are denoted by bold lowercase letters. Furthermore, given $N \in \mathbb{N}$, $\boldsymbol{I}_N$ denotes the identity matrix in $\mathbb{R}^{N \times N}$ and $\mathbb{I}(\cdot)$ denotes the indicator function.

### 2.1 Base arms and their outcomes, super arms and their rewards

Our setup involves base arms that are defined by their contexts, $x$, belonging to the context set $\mathcal{X}$. Let $r(x)$ represent the random outcome generated by the base arm with context $x$. We assume that there exists a function $f : \mathcal{X} \to \mathbb{R}$, such that we have $r(x) = f(x) + \eta$, for every $x \in \mathcal{X}$, where $\eta \sim \mathcal{N}(0, \sigma^2)$. We assume that $\eta$ is mutually independent across base arms and observations.

A super arm $S$ is a feasible subset of the base arm set and it is defined by the contexts of its base arms. Consider a super arm $S$ associated with the context tuple $\boldsymbol{x} = [x_1, \ldots, x_{|S|}]^T$ such that $x_m \in \mathcal{X}$, $\forall m \in \{1, \ldots, |S|\}$. The corresponding outcome and expected outcome vectors (the latter is also called the expectation vector) are denoted by $r(\boldsymbol{x}) = [r(x_1), \ldots, r(x_{|S|})]^T$ and $f(\boldsymbol{x}) = [f(x_1), \ldots, f(x_{|S|})]^T$. We assume that the reward received from playing this super arm is a non-negative random variable denoted by $U(r(\boldsymbol{x}))$. Moreover, we assume that the expected reward of playing any super arm is a function only of the set of arms and the mean vector, and we define $u(f(\boldsymbol{x})) = \mathbb{E}[U(r(\boldsymbol{x}))|f]$. Again, this assumption is standard in combinatorial bandits (Chen et al., 2013). In addition, we impose the following mild assumptions on $u$, which allow for a very large class of functions to fit our model such as multi-armed bandit with multiple plays, maximum weighted bipartite matching, and probabilistic maximum coverage (Chen et al., 2013).

**Assumption 1.** *(Monotonicity) Let $S$ be a feasible super arm. For any $f = [f_1, \ldots, f_{|S|}]^T \in \mathbb{R}^{|S|}$ and $f' = [f'_1, \ldots, f'_{|S|}]^T \in \mathbb{R}^{|S|}$, if $f_m \leq f'_m$, $\forall m \in \{1, \ldots, |S|\}$, then $u(f) \leq u(f')$.*

**Assumption 2.** *(Lipschitz continuity of the expected reward in expected outcomes) $\exists B > 0$ such that for any feasible super arm $S$, $f = [f_1, \ldots, f_{|S|}]^T \in \mathbb{R}^{|S|}$ and $f' = [f'_1, \ldots, f'_{|S|}]^T \in \mathbb{R}^{|S|}$, we have $|u(f) - u(f')| \leq B \sum_{i=1}^{|S|} |f_i - f'_i|$.*

Assumption 1 states that the expected reward is monotonically non-decreasing with respect to the expected outcome vector. Assumption 2 implies that the expected reward varies smoothly as a function of expected base arm outcomes.

### 2.2 The optimization and learning problems

We consider a sequential decision-making problem with time-varying base arms that proceeds over $T$ rounds indexed by $t \in [T]$. The agent knows $u$ perfectly but does not know $f$ beforehand. In each round $t$, $M_t$ base arms indexed by the set $\mathcal{M}_t = [M_t]$ are available. We assume $\max_{t \geq 1} M_t \leq M$, for some integer $M$. The context of base arm $m \in \mathcal{M}_t$ is represented by $x_{t,m} \in \mathcal{X}$. We denote by $\mathcal{X}_t = \{x_{t,m}\}_{m \in \mathcal{M}_t}$ the set of available contexts and by $f_t = [f(x_{t,m})]_{m \in \mathcal{M}_t}^T$ the vector of expected outcomes of the available base arms in round $t$. We denote by $\mathcal{S}_t$ the set of feasible super arms in round $t$ and $\mathcal{S} = \cup_{t \geq 1} \mathcal{S}_t$ the overall feasible set of super arms. Note that $\mathcal{S}_t$ depends on the structure of the optimization problem which is known by the learner, and if $S \in \mathcal{S}_t$, then $S \subseteq \mathcal{M}_t$. Furthermore, we assume that the budget (maximum number of base arms in a super arm) does not exceed some fixed integer $K \in \mathbb{N}$, that is, for any $S \in \mathcal{S}$, we have $|S| \leq K$. At the beginning of round $t$, the agent first observes $\mathcal{M}_t$ and $\mathcal{X}_t$. Then, it selects a super arm $S_t$ from $\mathcal{S}_t$.

#### 2.2.1 The optimization problem

For a moment, consider the hypothetical situation where $f$ is known beforehand. Then the agent would have selected an optimal super arm given by $S_t^* \in \text{argmax}_{S \in \mathcal{S}_t} u(f(\boldsymbol{x}_{t,S}))$, and subsequently obtained an expected reward of $\text{opt}(f_t) = \max_{S \in \mathcal{S}_t} u(f(\boldsymbol{x}_{t,S}))$. However, it is known that, in general, combinatorial optimization is NP-hard (Wolsey & Nemhauser, 2014), and thus, efficient computation of $S_t^*$ is not possible, even if $f$ were known. Fortunately, for many combinatorial optimization problems of interest, there exist computationally efficient approximation oracles. Thus, we assume that the agent obtains $S_t$ from an $\alpha$-approximation oracle, which when given as input $f_t$ and the specific structure of the particular optimization problem, returns a super arm $\text{Oracle}(f_t)$[1] such that $u(f(\boldsymbol{x}_{t,\text{Oracle}(f_t)})) \geq \alpha \times \text{opt}(f_t)$.

---

[1] Note that Oracle also takes as input the feasible set $\mathcal{S}_t$, but this is omitted from the notation for brevity.

### 2.2.2 The learning problem

Since the agent does not know $f$ in our case, it calls the approximation oracle in each round $t$ with an $M_t$-dimensional parameter vector $\theta_t$ to get $S_t = \text{Oracle}(\theta_t)$. Here, $\theta_t$ is an approximation to $f_t$ calculated based on the history of observations. We will show in Section 3 that our learning algorithm uses upper confidence bounds (UCBs) based on GP posterior mean and variances as $\theta_t$. It is important to note here that $S_t$ is an $\alpha$-optimal solution under $\theta_t$ but not necessarily under $f_t$. At the end of round $t$, the agent collects the reward $U(r(\boldsymbol{x}_{t,S_t}))$ where $\boldsymbol{x}_{t,S_t} = [x_{t,s_{t,1}}, \ldots, x_{t,s_{t,|S_t|}}]^T$ is the set of context vectors associated with super arm $S_t$ and $r(\boldsymbol{x}_{t,S_t}) = [r(x_{t,s_{t,1}}), \ldots, r(x_{t,s_{t,|S_t|}})]^T$ is the outcome vector of super arm $S_t$. It also observes $r(\boldsymbol{x}_{t,S_t})$ as a part of the semi-bandit feedback. The goal of the agent is to maximize its expected cumulative reward in the long run.

To measure the loss of the agent in this setting by round $T$ for a given sequence of base arm availability $\{\mathcal{X}_t\}_{t=1}^T$, we use the standard notion of $\alpha$-approximation regret (referred to as the regret hereafter), which is given as $R_\alpha(T) = \alpha \sum_{t=1}^T \text{opt}(f_t) - \sum_{t=1}^T u(f(\boldsymbol{x}_{t,S_t}))$.

### 2.3 Example applications of C3-MAB

In this section, we detail some applications of C3-MAB.

**Dynamic maximum weighted bipartite matching.** Let $G_t = (L_t, R, E_t)$ represent the bipartite graph in round $t$, where $L_t$ and $R$ are the set of nodes that form the parts of the graph such that $|L_t| \geq K$ and $|R| = K$ (here, $K$ represents the budget), and $E_t$ is the set of edges. Here, each edge $(i, j) \in E_t$, such that $i \in L_t$ and $j \in R$, represents a base arm. The weight of the edge $(i, j) \in E_t$ is given by $f(x_{t,(i,j)})$. In this problem, $\mathcal{S}_t$ corresponds to the set of $K$-element matchings of $G_t$. Hence, given $S \in \mathcal{S}_t$, $U(r(\boldsymbol{x}_{t,S})) = \sum_{m \in S} r(x_{t,m})$ and $u(f(\boldsymbol{x}_{t,S})) = \sum_{m \in S} f(x_{t,m})$. $S_t^*$ can be computed by the Hungarian algorithm or its variants (Kuhn, 1955) in polynomial computation time, and hence $\alpha = 1$. This problem can model the dynamic assignment of $|R|$ resources to $|L_t|$ available users in round $t$, where each resource can be assigned to at most one user. Anexample application is multi-user multi-channel communication (Gai et al., 2012).

**Dynamic probabilistic maximum coverage.** Let $G_t = (U_t, V_t, E_t)$ represent the bipartite graph in round $t$, where $U_t$ and $V_t$ are the set of nodes that form the parts of the graph, and $E_t$ is the set of edges such that $|U_t| \geq K$. Each edge $(i, j) \in E_t$ has a context-dependent activation probability given as $f(x_{t,(i,j)})$, i.e., when a node $i \in U_t$ is chosen, it activates its neighbor $j$ with probability $f(x_{t,(i,j)})$ independent of the other neighbors of $j$. Here, the goal is to choose a subset of $K$ nodes in $U_t$ that maximizes the expected number of activated nodes in $V_t$. Although this problem is known to be NP-hard (Chen et al., 2016a), there exists a deterministic $\alpha = (1 - 1/e)$-approximation oracle (Nemhauser et al., 1978).

### 2.4 Putting structure on base arm outcomes via GPs

Since the agent does not have any control over $M_t$ and $\mathcal{X}_t$, the expected outcomes of available base arms can vary greatly over the rounds. Since how well the learner performs depends on how well it learns the unknown $f$, we need to impose regularity conditions on $f$. In this paper, we model $f$ as a sample from a Gaussian process, which is defined below.

**Definition 1.** *A Gaussian Process with index set $\mathcal{X}$ is a collection $(f(x))_{x \in \mathcal{X}}$ of random variables which satisfies the property that $(f(x_1), \ldots, f(x_n))$ is a Gaussian random vector for all $\{x_1, \ldots, x_n\} \in \mathcal{X}$ and $n \in \mathbb{N}$. The probability law of a GP $(f(x))_{x \in \mathcal{X}}$ is uniquely specified by its mean function $x \mapsto \mu(x) = \mathbb{E}[f(x)]$ and its covariance function $(x_1, x_2) \mapsto k(x_1, x_2) = \mathbb{E}[(f(x_1) - \mu(x_1))(f(x_2) - \mu(x_2))]$.*

We assume that for every $x \in \mathcal{X}$, we have $k(x, x) \leq 1$. This is a general assumption that is widely used in the GP bandits literature (Srinivas et al., 2012). Now let us recall the closed-form expressions for the posterior distribution of GP-sampled functions, given a set of observations. Fix $N \in \mathbb{N}$. Consider a finite sequence $\boldsymbol{x}_{[N]} = [x_1, \ldots, x_N]^T$ of contexts with the corresponding vector $\boldsymbol{r}_{[N]} := r(\boldsymbol{x}_{[N]}) = [r(x_1), \ldots, r(x_N)]^T$ of

outcomes and corresponding vector $\boldsymbol{f}_{[N]} = [f(x_1), \ldots, f(x_N)]^T$ of expected outcomes. For every $n \leq N$, we have $r(x_n) = f(x_n) + \eta_n$, where $\eta_n$ is the noise that corresponds to this particular outcome.

The posterior distribution of $f$ given $r_{[N]}$ is that of a GP with mean function $\mu_N$ and covariance function $k_N$ given by

$$\mu_N(x) = (k_{[N]}(x))^T (\boldsymbol{K}_{[N]} + \sigma^2 \boldsymbol{I}_N)^{-1} r_{[N]}, \tag{1}$$

$$k_N(x, x') = k(x, x') - (k_{[N]}(x))^T (\boldsymbol{K}_{[N]} + \sigma^2 \boldsymbol{I}_N)^{-1} k_{[N]}(x'), \tag{2}$$

$$\sigma_N^2(x) = k_N(x, x), \tag{3}$$

where $k_{[N]}(x) = [k(x_1, x), \ldots, k(x_N, x)]^T \in \mathbb{R}^{N \times 1}$ and $\boldsymbol{K}_{[N]} = [k(x_i, x_j)]_{i=1, j=1}^N$. In particular, the posterior distribution of $f(x)$ is $\mathcal{N}(\mu_N(x), \sigma_N^2(x))$ and that of the corresponding observation $r(x)$ is $\mathcal{N}(\mu_N(x), \sigma_N^2(x) + \sigma^2)$.

### 2.5 Information gain

In Bayesian optimization, the informativeness of a finite sequence $\boldsymbol{x}_{[N]}$ is quantified by the information gain, defined as $I(\boldsymbol{r}_{[N]}; \boldsymbol{f}_{[N]}) = H(\boldsymbol{r}_{[N]}) - H(\boldsymbol{r}_{[N]} | \boldsymbol{f}_{[N]})$, where $H(\cdot)$ denotes the entropy of a random variable and $H(\cdot | \boldsymbol{f}_{[N]})$ denotes the conditional entropy of a random variable with respect to $\boldsymbol{f}_{[N]}$. The information gain gives us the decrease in entropy of $\boldsymbol{f}_{[N]}$ given the outcomes $\boldsymbol{r}_{[N]}$. We define the maximum information gain as $\gamma_N = \max_{\boldsymbol{x}_{[N]}} I(\boldsymbol{r}_{[N]}; \boldsymbol{f}_{[N]})$.

Note that $\gamma_N$ is the maximum information gain associated with any $N$-tuple of elements of $\mathcal{X}$. For large (potentially infinite) spaces this quantity can be very large, depending on the chosen kernel. On the other hand, the dynamically varying base arm availability in our setting does not require such information. We only need the maximum possible information coming from a fixed sequence of base arm (context) arrivals. Thus, such a notion becomes redundant in this case. This motivates us to relate the growth rate of the regret with the information content of the available base arms. Hence, we begin by adapting the definition of the maximum information gain so that it accommodates base arm availability, and guarantees the tightness of the regret bounds. Given $t \geq 1$, let $\mathcal{Z}_t \subset 2^{\mathcal{X}_t}$ be the set of context vectors corresponding to the feasible set $\mathcal{S}_t$ of super arms. Let $\boldsymbol{z}_t := \boldsymbol{x}_{t,S}$, for some $S \in \mathcal{S}_t$, be an element of $\mathcal{Z}_t$ and let $\boldsymbol{z}_{[T]} = [\boldsymbol{z}_1^T, \ldots, \boldsymbol{z}_T^T]$. We define the maximum information gain associated with the sequence of context arrivals $\mathcal{X}_1, \ldots, \mathcal{X}_T$ as

$$\gamma_{KT}(\mathcal{X}_{\overline{T}}) = \max_{\boldsymbol{z}_{[T]} : \boldsymbol{z}_t \in \mathcal{Z}_t, t \leq T} I(r(\boldsymbol{z}_{[T]}); f(\boldsymbol{z}_{[T]})) , \tag{4}$$

where $\mathcal{X}_{\overline{T}} := \cup_{t \leq T} \mathcal{X}_t$. The information gain from any sequence of $T$ selected super arms is upper bounded by $\gamma_{KT}(\mathcal{X}_{\overline{T}})$. Our regret bounds depend on it. Note that $\gamma_{KT}(\mathcal{X}_{\overline{T}})$ is the usual notion of the maximum information gain applied onto the union of all available context sets over $T$ rounds.

## 3 The learning algorithm

Our algorithm is called *Optimistic Combinatorial Learning and Optimization with Kernel Upper Confidence Bounds* (O'CLOK-UCB), with pseudocode given in Algorithm 1. The procedure is described as follows: At round $t$, we observe available base arms and their contexts. For each available base arm, we maintain an index which is an upper confidence bound on its expected outcome. Let $S_1, \ldots, S_{t-1}$ be a sequence of super arms. Given base arm $m$ with its associated context $x_{t,m}$, we define its index based on the observations $[\boldsymbol{r}_{S_1}^T, \ldots, \boldsymbol{r}_{S_{t-1}}^T]$ as:

$$i_t(x_{t,m}) = \mu_{[\![t-1]\!]}(x_{t,m}) + \beta_t^{1/2} \sigma_{[\![t-1]\!]}(x_{t,m}) , \tag{5}$$

where $\mu_{[\![t-1]\!]}(\cdot)$ and $\sigma_{[\![t-1]\!]}^2(\cdot)$ stand for the posterior mean and variance (respectively) given the vector $[\boldsymbol{r}_{S_1}^T, \ldots, \boldsymbol{r}_{S_{t-1}}^T]$ of observations up to round $t$. Furthermore, $\beta_t$ is a parameter that depends on $t$ whose exact value will be specified later. We denote by $i_t(\boldsymbol{x}_{t,S_t})$ the vector $[i_t(x_{t,s_{t,1}}), \ldots, i_t(x_{t,s_{t,|S_t|}})]^T$. Note that our definition of the index of an arm with context $x$ in the beginning of round $t$ depends on $\mu_{[\![t-1]\!]}(x)$ and

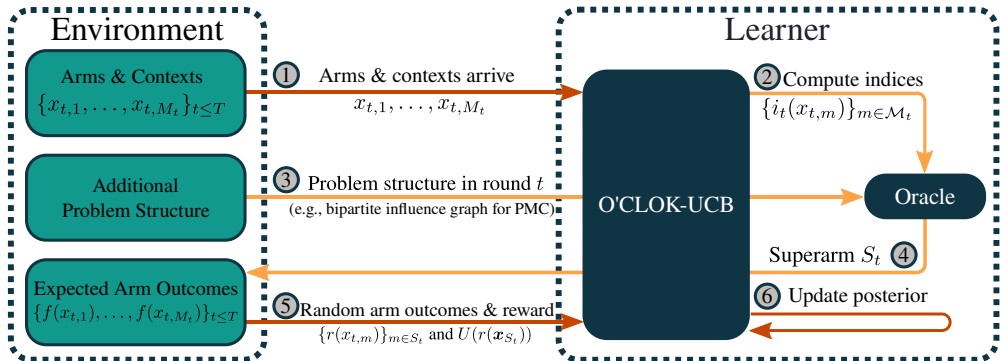

Figure 1: Illustration of the steps of our algorithm for round $t$. PMC stands for probabilistic maximum coverage.

$\sigma_{[\![t-1]\!]}(x)$. This is due to the combinatorial nature of the problem. Since, at the beginning of round $t$, the algorithm has obtained semi-bandit feedback from $\sum_{t'=1}^{t-1} |S_{t'}|$ context selections, the posterior mean and variance of $f$ calculated at that round will depend on the selected super arms, namely, $S_1, \ldots, S_{t-1}$.

---

**Algorithm 1** O'CLOK-UCB

---

**Input:** $\mathcal{X}$, $K$, $M$; GP prior: $\mu_0 = \mu$, $k_0 = k$.
  **Initialize**: $\mu_0 = \mu$, $k_0 = k$.
  **for** $t = 1, \ldots, T$ **do**
      Observe base arms in $\mathcal{M}_t$ and their contexts $\mathcal{X}_t$.
      **for** $x_{t,m} : m \in \mathcal{M}_t$ **do**
          Calculate $\mu_{[\![t-1]\!]}(x_{t,m})$ and $\sigma_{[\![t-1]\!]}(x_{t,m})$ as in equation 1 and equation 3.
          Compute index $i_t(x_{t,m})$ as in equation 5.
      **end for**
      $S_t \leftarrow \text{Oracle}(i_t(x_{t,1}), \ldots, i_t(x_{t,M_t}))$.
      Observe outcomes of base arms in $S_t$ and collect the reward.
  **end for**

---

After the indices of the available base arms (i.e., $\{i_t(x_{t,m})\}_{m \in \mathcal{M}_t}$) are computed, they are given as input $\theta_t$ to the approximation oracle in round $t$ to obtain the super arm $S_t = (s_{t,1}, \ldots, s_{t,|S_t|}) \subseteq \mathcal{M}_t$ that will be played in round $t$. Note that $S_t$ is an (approximately) optimal solution under $\theta_t$, but not necessarily under $f_t$. We assume a deterministic oracle in order to guarantee our theoretical results.

After observing the semi-bandit feedback from the selected arms, at the beginning of the next round, we update the posterior distribution of $f$ based on $S_t$ according to rules equation 1 and equation 3, which will be used to compute the indices for the next round. Figure 1 illustrates the steps of our algorithm for round $t$.

## 4 Regret bounds

We start by stating our main result which gives a high probability upper bound on the $\alpha$-regret in terms of $\gamma_{KT}(\mathcal{X}_{\overline{T}})$. Detailed proofs of all the stated results can be found in Appendix A.

**Theorem 1.** *Let $\delta \in (0,1)$, $T \in \mathbb{N}$. Given $\beta_t = 2\log(M\pi^2 t^2/3\delta)$, the regret incurred by O'CLOK-UCB in $T$ rounds is upper bounded as $R_\alpha(T) \leq \sqrt{C(K)K\beta_T T \gamma_{KT}(\mathcal{X}_{\overline{T}})}$, with probability at least $1 - \delta$, where $C(K) = 8B^2(\lambda^*(K) + \sigma^2)$, and $\lambda^*(K)$ is the maximum eigenvalue of all covariance matrices of selected actions over $T$ rounds.*

To prove Theorem 1, we need to utilize a new result that gives lower bounds on $\gamma_{KT}(\mathcal{X}_{\overline{T}})$ in terms of quantities that are similar to those that upper bound the expected regret.

**Lemma 1.** *Let $\boldsymbol{z}_t := \boldsymbol{x}_{t,S_t}$ be the vector of selected contexts at time $t \geq 1$. Given $T \geq 1$, we have:*

$$I\left(r(\boldsymbol{z}_{[T]}); f(\boldsymbol{z}_{[T]})\right) \geq \frac{1}{2(\sigma^{-2}\lambda^*(K)+1)} \sum_{t=1}^{T} \sum_{k=1}^{|S_t|} \sigma^{-2}\sigma_{[\![t-1]\!]}^2(\overline{x}_{t,k}) ,$$

*where $\boldsymbol{z}_{[T]} = [\boldsymbol{z}_1, \ldots, \boldsymbol{z}_T]^T$ is the vector of all selected contexts until round $T$ and $\lambda^*(K)$ is the maximum eigenvalue of matrices $(\Sigma_{[\![t-1]\!]}(\boldsymbol{z}_{[t]}))_{t=1}^{T}$.*

**Remark 1.** *Since, in every round, the outcomes of the selected base arms are observed only after the selection process is over, we cannot reduce the problem to a sequential decision-making in $KT$ rounds, in which case, the outcomes would allow for the sequential update of the indices and let us directly use the formula of the maximum information gain from (Srinivas et al., 2012). We solve this problem by lower bounding the information gain. Consequently, Lemma 1 takes into account the contextual combinatorial nature of the problem.*

Finally, using kernel-dependent explicit bounds on $\gamma_{KT}(\mathcal{X}_{\overline{T}})$ given in (Srinivas et al., 2012; Vakili et al., 2020), we state a corollary of Theorem 1 which gives similar bounds on the $\alpha$-regret incurred by O'CLOK-UCB.

**Corollary 1.** *Let $\delta \in (0,1)$, $T, K \in \mathbb{N}$ and let $\mathcal{X} \subset \mathbb{R}^D$ be compact and convex. Under the conditions of Theorem 1 and for the following kernels, the $\alpha$-regret incurred by O'CLOK-UCB in $T$ rounds is upper bounded (up to polylog factors) with probability at least $1 - \delta$ as follows:*

- *For the linear kernel we have: $R_\alpha(T) \leq \tilde{O}\left(\sqrt{\lambda^*(K)DKT}\right)$.*

- *For the RBF kernel we have: $R_\alpha(T) \leq \tilde{O}\left(\sqrt{\lambda^*(K)KT\log^D T}\right)$.*

- *For the Matérn kernel we have: $R_\alpha(T) \leq \tilde{O}\left(\sqrt{\lambda^*(K)K}T^{(D+\nu)/(D+2\nu)}\right)$, where $\nu > 1$ is the Matérn parameter.*

## 5 Experimental results

We perform simulations on: (i) a crowdsourcing setup with computational time comparisons, (ii) a synthetic dataset to show how our algorithm exploits arm dependence, and (iii) a real-world movie recommendation setup. All simulations were run using Python, with source code provided on our GitHub,[2] on a PC running Ubuntu 16.04 LTS with an Intel Core i7-6800K CPU, an Nvidia GTX 1080Ti GPU, and 32 GB of 2133 MHz DDR4 RAM.

### 5.1 Simulation I: Crowdsourcing

We evaluate the performance of O'CLOK-UCB by comparing it with ACC-UCB (Nika et al., 2020) and AOM-MC (Chen et al., 2018), two UCB-based algorithms. We perform semi-synthetic crowdsourcing simulations using the Foursquare dataset. The Foursquare dataset (Yang et al., 2015) contains check-in data from New York City (227,428 check-ins) and Tokyo (573,703 check-ins) for a period of 10 months from April 2012 to February 2013. Each check-in comes with a location tag as well as the time of the check-in. In our simulations, we use the TKY dataset because its locations are more spread out than the NYC dataset.

### 5.1.1 Simulation Setup

We perform a crowdsourcing simulation where the goal is to assign workers to arriving tasks. In our simulation, 250 tasks arrive (i.e., $T = 250$), each with a location (normalized longitude and latitude) sampled

---

[2]Link: https://github.com/Bilkent-CYBORG/OCLOK-UCB

uniformly at random from $[0,1]^2$ and a difficulty rating, sampled from $[0,1]$ (0 is most difficult). Similarly, each worker also has a 2D location in $[0,1]^2$, sampled from the TKY dataset, and a battery status, sampled from $[0,1]$.

In round $t$, the agent observes the set of available workers, the expected number of which is sampled from a Poisson distribution with mean 100. The set is composed of workers whose distance (Euclidean norm) to the task is less than $\sqrt{0.5}$. Then, each worker-task pair is an arm, with a three-dimensional context comprised of the scaled distance between the worker and task,[3] the task's difficulty, and the worker's battery. We also set a fixed budget constraint of $K = 5$, which means that 5 workers should be chosen for each task.

Then, given the task-worker joint context (i.e., arm context) $x$ and, noting that $x^1, x^2$, and $x^3$ represent the worker-task distance, task difficulty, and worker battery, respectively, we define the expected outcome of each base arm as $f(x) = \mathbb{E}[r(x)] = Ag\left(x^1\right)\sqrt{x^2 \cdot x^3}$, where $g$ is a Gaussian probability density function with mean 0 and standard deviation 0.4, and $A = \sqrt{2\pi \cdot 0.4^2}$ is the scaling constant. Note that $f$ is decreasing in the worker-task distance and task difficulty, and increasing in the worker's battery. Then, the random quality of each worker with joint context $x$ is defined as $r(x) = f(x) + \eta$, where $\eta$ is zero mean noise with a standard deviation of 0.1.

Finally, in round $t$ and given $K$ chosen workers with joint task-worker context vector $\boldsymbol{x} = [x_1, \ldots, x_K]$, we define our reward as $U(r(\boldsymbol{x})) = \log\left(1 + \sum_{i=1}^{K} r(x_i)\right)$. Notice that the log term reflects the diminishing return on having multiple workers. Moreover, since log is an increasing function, the Oracle for this setup will be a greedy one that picks the arms whose contexts have the highest performance estimates.

### 5.1.2 Algorithms

**O'CLOK-UCB**: We use the GPflow library (Matthews et al., 2017) for the sampling from and updating the Gaussian Process. We set $\delta = 0.05$ and use a squared exponential kernel with both variance and lengthscale set to 1.

**SO'CLOK-UCB**: In this variation of our algorithm, we use the sparse approximation to the GP posterior described in (Titsias, 2009). In this sparse approximation, instead of using all of the arm contexts up to round $t$ to compute the posterior, a small $s$ element subset of them is used, called the inducing points. The non-sparse O'CLOK-UCB requires updating the GP posterior at each round. A standard, efficient implementation performs this sequentially using block matrix inversion, resulting in a total computational complexity of $O(K^3T^3)$ over $T$ rounds. By using a sparse approximation with $s$ inducing points, this is dramatically reduced to a total complexity of $O(s^2KT^2)$. We use this sparse variation with inducing points uniformly at random from all of the contexts picked so far. In other words, in round $t$, we sample $s$ contexts from $\{x_{1,1}, \ldots x_{1,K}, \ldots x_{t-1,K}\}$. We run simulations with three different inducing points: 10, 20, and 50. We also set $\delta = 0.05$ and use the squared exponential kernel.

**ACC-UCB**: We set $v_1 = \sqrt{3}, v_2 = 1, \rho = 0.5$, and $N = 8$, as given in Definition 1 of (Nika et al., 2020). The initial (root) context cell, $X_{0,1}$, is a three dimensional unit hypercube centered at $(0.5, 0.5, 0.5)$.

**CC-MAB**: Since we have a three-dimensional context space, 300 tasks (rounds), and an exact Oracle, we set $h_T = \lceil 300^{\frac{1}{3 \cdot 1 + 3}} \rceil = 3$, as given in Theorem 1 of (Chen et al., 2018). Hence, each hypercube has a length of $1/h_T = \frac{1}{3}$.

**Benchmark**: The benchmark greedily picks the arms with the highest expected outcome.

---

[3]To scale the distance we divide it by $\sqrt{0.5}$ because that is the maximum possible distance.

### 5.1.3 Results

We ran 5 independent runs and averaged the reward, regret, and time taken over these runs. We present the standard deviations of the runs as error bars. Figure 2 shows the average task reward up to task $t$, divided by the benchmark reward; and Figure 3 shows the cumulative regret of each algorithm for different $T$. The two UCB-based algorithms perform around the same and are equal in performance with SO'CLOK-UCB with 10 inducing points. However, they perform substantially worse for a larger number of inducing points. Interestingly, only 100 inducing points were enough to achieve a very close performance to O'CLOK-UCB, which does not use any approximations for the posterior update. Finally, as expected, the smaller the number of inducing points, the worse the performance of SO'CLOK-UCB compared with the non-sparse O'CLOK-UCB algorithm. However, even with 20 inducing points, SO'CLOK-UCB is able to outperform both CC-MAB and ACC-UCB by around 30%.

We also plot the time taken for each algorithm to process each round, given in Figure 4. We can see that the running time of O'CLOK-UCB does not scale well with the number of rounds, taking more than 20 minutes just to pick workers and update the posterior for the final task. On the other hand, the sparse algorithms do much better and are not only four to five times faster, but also scale linearly with $t$. The latter, coupled with the fact that SO'CLOK-UCB's performance is practically identical to that of O'CLOK-UCB, means that even though SO'CLOK-UCB is slower than the UCB-based algorithms, its performance scales well with time and so it can be used for large simulations or online settings. It is worth noting that in Figure 4, the runtime curves for ACC-UCB and CC-MAB are superimposed, reflecting their similar and highly efficient computational performance compared to the GP-based methods.

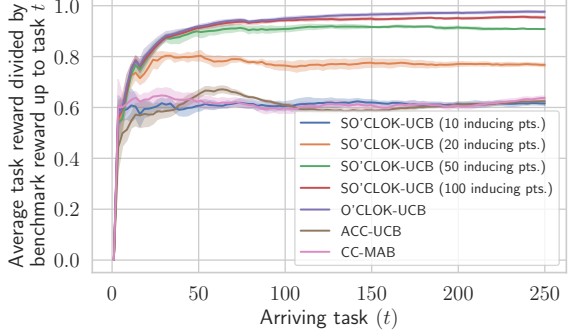
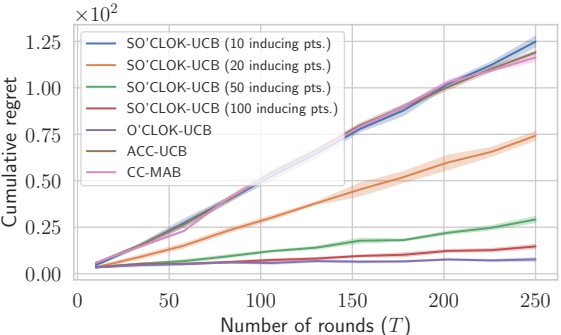

Figure 2: Average reward of each algorithm divided by that of the benchmark Simulation I.

Figure 3: Cumulative regret of each algorithm for different number of rounds ($T$) Simulation I.

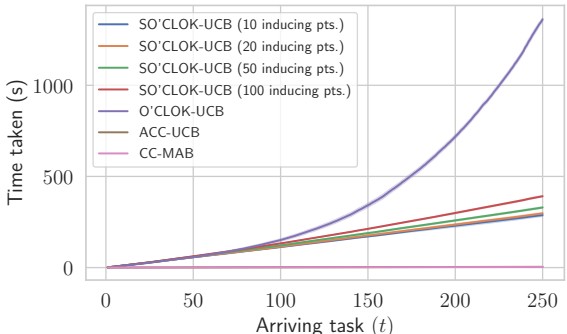
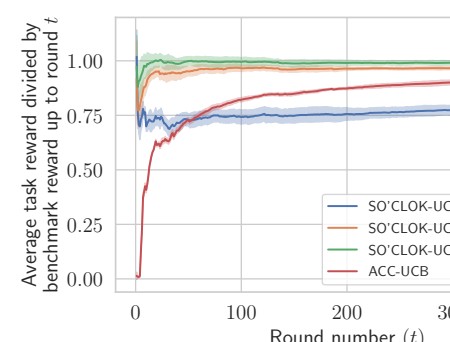

Figure 4: Time taken in seconds to process each round for each algorithm in Simulation I.

Figure 5: Average reward of each algorithm divided by that of the benchmark for the Movie-Lens DPMC setup of Simulation III.

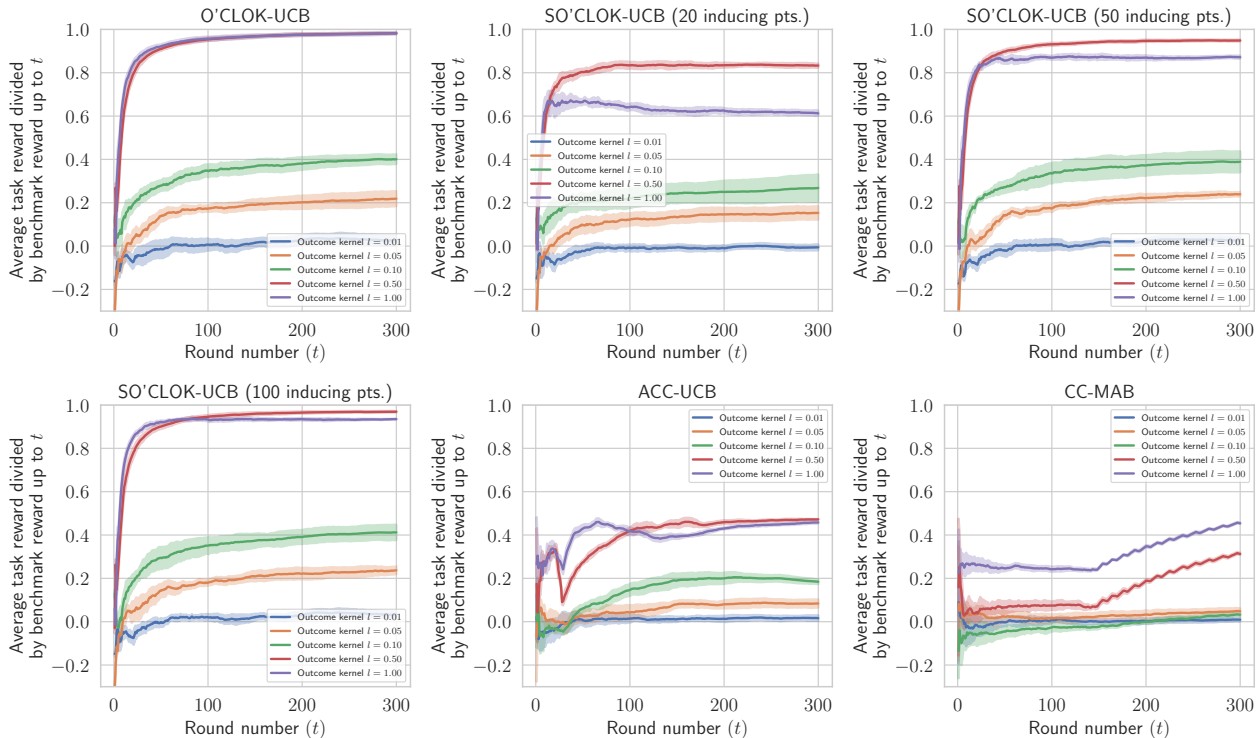

Figure 6: Average reward of each algorithm divided by that of the benchmark for different outcome kernel lengthscales ($l$) in Simulation II.

## 5.2 Simulation II: Varying Base Arm Codependency

The goal of this simulation is to evaluate the performance of our approach against baseline methods under varying levels of base arm codependency, demonstrating how our GP-based method effectively exploits base arm dependencies.

### 5.2.1 Setup

We perform 30 different simulations with an evenly spaced number of rounds from $T = 10$ to $T = 300$ in order to be able to plot the cumulative regret. Note that this step is needed because ACC-UCB and CC-MAB's decisions are affected by the number of rounds, hence to plot the cumulative regret, we need to run the simulations with different $T$.

Similar to Simulation I, the number of arms in each round is sampled from a Poisson distribution with a mean of 100 and the budget is $K = 5$. This means that we will have 30000 base arms when $T = 300$. Sampling this many points from a GP requires the creation and Cholesky decomposition of a 900 million element matrix, which would not only take a long time on our i7 6700K PC, but would also require a large amount of RAM (more than 7.2 GB). In order to address this issue, we first generate 6000 3D contexts from $[0, 1]^3$ and then sample the GP at those points. Then, during our simulation, we sample each base arm's context $x$ and the corresponding expected outcome $f(x)$ from the generated sets. Just like in Simulation I, $r(x) = f(x) + \eta$, where $\eta \sim \mathcal{N}(0, 0.1^2)$. Note that we first generate the dataset (i.e., arriving arm contexts and rewards in each round) for $T = 300$ and then use the same dataset, but truncated, for smaller $T$.

The expected outcome of each arm is generated from a zero mean GP with the squared exponential kernel with lengthscale $l$, given as $k(x, x') = \exp\left(-\frac{1}{2l^2}\|x - x'\|\right)$. We run our simulations for five different lengthscales, $l \in \{0.01, 0.05, 0.1, 0.5, 1\}$. Intuitively, the larger the length scale, the larger the covariance (and thus dependence) between any two arms with nonequal contexts.

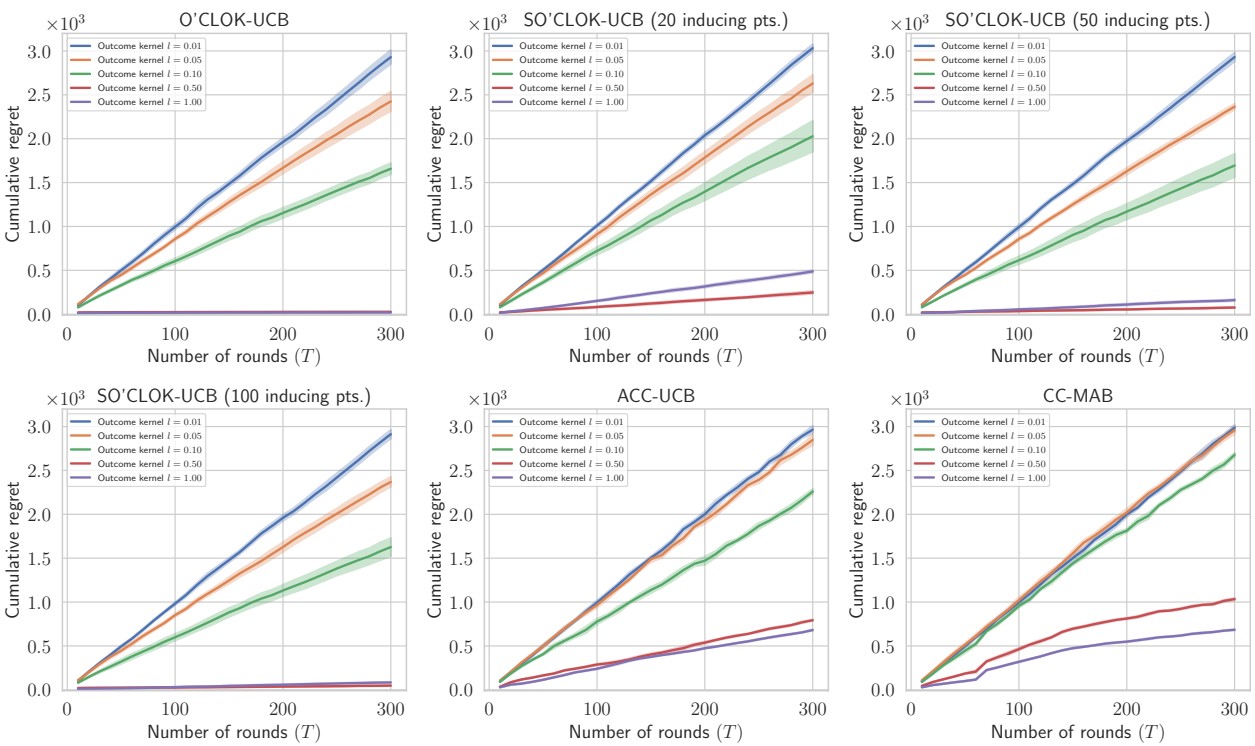

Figure 7: Cumulative regret of each algorithm at the end of different number of rounds and outcome kernel lengthscales ($l$) in Simulation II.

Finally, in round $t$ and given $K$ chosen arms context vector $\boldsymbol{x} = [x_1, \ldots, x_K]$, we define our reward as $u(r(\boldsymbol{x})) = \sum_{i=1}^{K} r(x_i)$. Just like in Simulation I, each algorithm will select five arms (i.e., $K = 5$). Moreover, the Oracle for this setup will be a greedy one that picks the arms whose contexts have the highest performance estimates.

### 5.2.2 Algorithms

We use the same algorithms as in Simulation I, detailed below.

**O'CLOK-UCB**: We use the same squared exponential kernel as in Simulation I and set $\delta = 0.05$.

**SO'CLOK-UCB**: We run simulations with three different inducing points: 20, 50, and 100. We also set $\delta = 0.05$ and use the same squared exponential kernel as O'CLOK-UCB.

**ACC-UCB**: We set $v_1 = \sqrt{3}, v_2 = 1, \rho = 0.5$, and $N = 8$. The initial (root) context cell, $X_{0,1}$, is a three dimensional unit hypercube centered at $(0.5, 0.5, 0.5)$.

**CC-MAB**: Since we have a three-dimensional context space, 300 tasks (rounds), and an exact Oracle, we set $h_T = \lceil 300^{\frac{1}{3 \cdot 1 + 3}} \rceil = 3$. Hence, each hypercube has a length of $1/h_T = \frac{1}{3}$.

**Benchmark**: The benchmark greedily picks the arms with the highest expected outcome.

### 5.2.3 Results

We run each simulation five times and average over them to get the results. We also visualize the standard deviation of the runs as error bars. Figure 6 shows the average task reward up to task $t$, divided by the benchmark reward; and Figure 7 shows the final cumulative regret of each algorithm for different $T$. None of the algorithms is able to learn and achieve high reward when $l = 0.01$, which is expected because different arm outcomes are almost independent of one another for this length scale. As the length scale increases, we see that all algorithms improve, but it is only the GP algorithms that manage to achieve performance close to that of the benchmark (i.e., approach $> 0.8$ in reward plots). SO'CLOK-UCB with 20 inducing points performed the worst among the GP algorithms, which is expected because it is essentially trying to learn the entire $[0,1]^3$ arm context space with just 20 randomly picked contexts. However, when the number of inducing points increases, the performance of SO'CLOK-UCB with both 50 and 100 inducing points is practically identical to the non-sparse O'CLOK-UCB's performance when the outcome kernel has $l = 1$. Interestingly, O'CLOK-UCB achieves near-optimal performance, as shown by its scaled reward being very close to one, when the outcome kernel has $l = 0.5$ or $l = 1$, but the same cannot be said about SO'CLOK-UCB. Moreover, O'CLOK-UCB is the only GP algorithm that has better performance when $l = 1$ compared with $l = 0.5$. This indicates that the sparse algorithms are not able to make use of the increased arm outcome covariance, even with 100 inducing points. That being said, the difference in the average reward between O'CLOK-UCB and SO'CLOK-UCB with 100 inducing points is only about 5% when $l = 1$ and 1% when $l = 0.5$. Thus, SO'CLOK-UCB is a very good practical alternative to O'CLOK-UCB.

## 5.3 Simulation III: Movie Recommendation

### 5.3.1 Setup & dataset

We use a movie recommendation setup where in each round a selection of $K = 3$ movies are to be shown to a group of users, with the goal being maximizing the number of users that watch any recommended movie. We use the MovieLens 25M dataset (Harper & Konstan, 2015). Each movie comes with genre metadata that indicates the genres of the movie from a set of 20 genres (action, adventure, comedy, etc.). Moreover, each rating is between 0.5 and 5.0, increasing in increments of 0.5.

In our experiment, we only consider ratings after 2015 and users who have rated at least 200 movies. Then, in each round $t$, we sample $L_t$ movies from the movie set, where $L_t$ follows Poisson with mean 75. Then, from the users who reviewed the picked movies, we sample $R_t$ of them, where $R_t$ follows Poisson with mean 200. If a user $j$ rated a movie $i$, then there is an edge connecting them with context $x_{i,j}$. We define this context to be $x_{i,j} = \langle \boldsymbol{u_j}, \boldsymbol{g_i} \rangle / 10$, where $\boldsymbol{u_j}$ is the average of the genres of the movies that user $j$ rated, weighed by their rating, $\boldsymbol{g_i}$ is the genre vector of movie $i$, and 10 is a normalizing factor.[4] Then, we pick the expected outcome of a base arm (i.e., edge) with context $x_{i,j}$ to be $f(x_{i,j}) = 2/(1 + e^{-4x_{i,j}}) - 1$, to model a realistic non-linear relationship. Its S-shape captures a natural saturation effect, where the outcome is most sensitive to mid-range context values but has diminishing returns at the extremes. Then, the random outcome (i.e., the chance of the user watching the recommended movie) is a Bernoulli random variable with probability $f(x_{i,j})$. Finally, the reward is the number of users that watched at least one movie that they were recommended. Formally, the expected reward in round $t$ and given a super arm $S$ (i.e., the set of outgoing movie-user edges from the $K$ movies to be recommended) is

$$u(f(\boldsymbol{x}_{t,S})) = \sum_{j=1}^{R_t} \left( 1 - \prod_{i=1}^{L_t} \left( 1 - \mathbb{I}((i,j) \in S) f(x_{i,j}) \right) \right),$$

where $\mathbb{I}((i,j) \in S)$ indicates whether the edge connecting movie $i$ and user $j$ is among the picked base arms. Notice that even if the learner knew all the edge probabilities, the problem of picking left nodes (i.e., movies) to maximize the expected number of activated right nodes (i.e., users) is NP-hard and thus computationally intractable (Chen et al., 2016a). Therefore, we use an approximate oracle for both the learning algorithms and the benchmark. We chose TIM+ (Tang et al., 2014), which is an $(\alpha, \beta)$-approximate oracle with

---

[4]We divide by 10 and not 20 to normalize the context because the maximum number of genres that a movie has in the dataset is 10.

$\alpha = 1 - 1/e - \epsilon$ and $\beta = 1 - 3n^{-l}$, where $n$ is the total number of nodes. Although our setup assumes an $\alpha$-approximate oracle, our algorithm has no issue using an $(\alpha, \beta)$-approximate oracle in practice, as we will see in the results. Note that the oracle knows the problem structure and thus knows the number of left and right nodes as well as the edges connecting them, but it does not know the edge probabilities and instead takes them as input.

### 5.3.2 Algorithms

We run the experiment on the sparse version of our algorithm, SO'CLOK-UCB, and ACC-UCB of (Nika et al., 2020). We excluded our non-sparse algorithm because its computational efficiency is very poor, especially in practice, and as we will see, the sparse algorithm manages to perform extremely well anyways. Moreover, CC-MAB of (Chen et al., 2018) was excluded because its $(1 - 1/e)$-greedy oracle that approximately maximizes a reward function takes as input a set, but in the dynamic probabilistic maximum coverage (DPMC) problem, the input to the reward function is a vector. Below are the parameters and configurations of each used algorithm:

**SO'CLOK-UCB**: We use the sparse variation of our algorithm, as described in Simulation I. We use three different numbers of inducing points: 1, 2, and 4. We also set $\delta = 0.05$ and use the squared exponential kernel. We use the TIM+ oracle with $\epsilon = 0.1$ and $l = 1$.

**ACC-UCB**: We set $v_1 = 1, v_2 = 1, \rho = 0.5$, and $N = 2$, as given in Definition 1 of (Nika et al., 2020). The initial (root) context cell, $X_{0,1}$, is a one-dimensional unit hypercube (i.e., a line) centered at $(0.5)$. ACC-UCB also uses the TIM+ oracle with $\epsilon = 0.1$ and $l = 1$.

### 5.3.3 Results

We run the experiment for 400 rounds and repeat it 5 times, averaging over each run. Figure 5 shows the running average reward of each algorithm divided by that of the benchmark. Even with 2 inducing points, SO'CLOK-UCB manages to outperform ACC-UCB by around 5%. With 4 inducing points, SO'CLOK-UCB reaches an average reward that is only 0.5% less than that of the benchmark and outperforms ACC-UCB by more than 8%. These results show that SO'CLOK-UCB achieves optimal performance in a realistic setting where the underlying problem cannot be solved by greedily picking the base arms with the highest estimate outcomes.

## 6 Conclusion

We considered the contextual combinatorial multi-armed bandit with changing action set problem with semi-bandit feedback, where in each round, the agent has to play a feasible subset of the base arms in order to maximize the cumulative reward. Under the assumption that the expected base arm outcomes are drawn from a Gaussian process and that the expected reward is Lipschitz continuous with respect to the expected base arm outcomes, we proposed O'CLOK-UCB which incurs $\tilde{O}(\sqrt{\lambda^*(K)KT\gamma_{KT}(\cup_{t \leq T}\mathcal{X}_t)})$ regret in $T$ rounds. In experiments, we showed that sparse GPs can be used to speed up UCB computation without significantly degrading the performance. Our comparisons also indicated that GPs can transfer knowledge among contexts better than partitioning the contexts into groups of similar contexts based on a similarity metric. A potential limitation of our work is the assumption of access to offline approximation oracles and the Lipschitz continuity of the expected regret.

An interesting future research direction involves investigating how dependencies between base arms can be used for more efficient exploration. For instance, when the oracle selects base arms sequentially, it is possible to update the posterior variances of the not yet selected based arms by conditioning on the selected but not yet observed base arms. Another interesting direction would be to investigate how tight the dependence of the regret bounds is on $K$.

## Acknowledgments

This work was supported in part by the Scientific and Technological Research Council of Türkiye (TÜBİTAK) under Grants 215E342 and 124E065. The work of C. Tekin was supported by the BAGEP Award of the Science Academy; by the Turkish Academy of Sciences Distinguished Young Scientist Award Program (TÜBA-GEBİP-2023); by TÜBİTAK 2024 Incentive Award.

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

# Appendix

## A  Missing Proofs

The proof of Theorem 1 relies on the following auxiliary lemmas, which we state and prove hereon.

First, let us denote by $\boldsymbol{r}_{[\![t-1]\!]}$ the vector of observations made until the beginning of round $t$, that is,

$$\boldsymbol{r}_{[\![t-1]\!]} = [r^T(\boldsymbol{x}_{1,S_1}), \ldots, r^T(\boldsymbol{x}_{t-1,S_{t-1}})]^T \ .$$

For any $t \geq 1$, note that the posterior distribution of $f(x)$ given the observation vector $\boldsymbol{r}_{[\![t-1]\!]}$ is $\mathcal{N}(\mu_{[\![t-1]\!]}(x), \sigma^2_{[\![t-1]\!]}(x))$, for any $x \in \mathcal{X}_t$. Thus, applying the Gaussian tail bound, we obtain

$$\mathbb{P}\Big(|f(x) - \mu_{[\![t-1]\!]}(x)| > \beta_t^{1/2}\sigma_{[\![t-1]\!]}(x)\big|\boldsymbol{r}_{[\![t-1]\!]}\Big) \leq 2\exp\left(\frac{-\beta_t}{2}\right) \ , \tag{6}$$

for $\beta_t \geq 0$. We will use this argument in order to prove our first result which gives high probability upper bounds on the deviation from the function value of the index of any available context.

Our first result guarantees that the indices upper bound the expected base arm outcomes with high probability.

**Lemma 2.** *For any $\delta \in (0,1)$, the probability of the following event is at least $1 - \delta$:*

$$\mathcal{F} = \{\forall t \geq 1, \forall x \in \mathcal{X}_t : |f(x) - \mu_{[\![t-1]\!]}(x)| \leq \beta_t^{1/2}\sigma_{[\![t-1]\!]}(x)\} \ ,$$

*where $\beta_t = 2\log(M\pi^2 t^2/3\delta)$.*

*Proof.* We have:

$$1 - \mathbb{P}(\mathcal{F}) = \mathbb{E}\left[\mathbb{I}\Big(\exists t \geq 1, \exists x \in \mathcal{X}_t : |f(x) - \mu_{[\![t-1]\!]}(x)| > \beta_t^{1/2}\sigma_{[\![t-1]\!]}(x)\Big)\right] \tag{7}$$

$$\leq \mathbb{E}\left[\sum_{t\geq 1}\sum_{x\in\mathcal{X}_t}\mathbb{I}\Big(|f(x) - \mu_{[\![t-1]\!]}(x)| > \beta_t^{1/2}\sigma_{[\![t-1]\!]}(x)\Big)\right]$$

$$= \sum_{t\geq 1}\sum_{x\in\mathcal{X}_t}\mathbb{E}\left[\mathbb{E}\left[\mathbb{I}\Big(|f(x) - \mu_{[\![t-1]\!]}(x)| > \beta_t^{1/2}\sigma_{[\![t-1]\!]}(x)\Big)\Big|\boldsymbol{r}_{[\![t-1]\!]}\right]\right] \tag{8}$$

$$= \sum_{t\geq 1}\sum_{x\in\mathcal{X}_t}\mathbb{E}\left[\mathbb{P}\Big(|f(x) - \mu_{[\![t-1]\!]}(x)| > \beta_t^{1/2}\sigma_{[\![t-1]\!]}(x)\Big|\boldsymbol{r}_{[\![t-1]\!]}\Big)\right]$$

$$\leq \sum_{t\geq 1}\sum_{x\in\mathcal{X}_t} 2\exp\left(\frac{-\beta_t}{2}\right) \tag{9}$$

$$= 2M \sum_{t\geq 1} \frac{6\delta}{2M\pi^2 t^2}$$

$$= \delta\frac{6}{\pi^2}\sum_{t\geq 1} t^{-2} = \delta \ ,$$

where equation 7 follows from the fact that $\mathbb{P}(\mathcal{F}) = \mathbb{E}[\mathbb{I}(\mathcal{F})]$; equation 8 follows from the tower rule and the fact that the sets $\mathcal{X}_t$, $t \geq 1$ are fixed (thus there is no randomness from there); equation 9 follows from equation 6 and the rest follows from substituting $\beta_t$ and the fact that $\sum_{t\geq 1} t^{-2} = \pi^2/6$. □

Next, we upper bound the gap of a selected super arm in terms of the gaps of individual arms. From here on, unless otherwise stated, we denote by $\overline{x}_{t,k}$ the context $x_{t,s_{t,k}}$ associated with the *kth* selected arm $s_{t,k}$ at time $t$ for brevity.

**Lemma 3.** *Given round $t \geq 1$, let us denote by $S_t^* = \{s_{t,1}^*, \ldots, s_{t,|S_t^*|}^*\}$ the optimal super arm in round $t$. Then, the following holds under the event $\mathcal{F}$:*

$$\alpha \cdot u(f(\boldsymbol{x}_{t,S_t^*})) - u(f(\boldsymbol{x}_{t,S_t})) \leq 2B\beta_t^{1/2} \sum_{k=1}^{|S_t|} \left|\sigma_{[\![t-1]\!]}(\overline{x}_{t,k})\right| .$$

*Proof.* Let us first define $G_t = \operatorname{argmax}_{S \in \mathcal{S}_t} u(i_t(\boldsymbol{x}_{t,S}))$. Given that event $\mathcal{F}$ holds, we have:

$$\alpha \cdot u(f(\boldsymbol{x}_{t,S_t^*})) - u(f(\boldsymbol{x}_{t,S_t})) \leq \alpha \cdot u(i_t(\boldsymbol{x}_{t,S_t^*})) - u(f(\boldsymbol{x}_{t,S_t})) \tag{10}$$

$$\leq \alpha \cdot u(i_t(\boldsymbol{x}_{t,G_t})) - u(f(\boldsymbol{x}_{t,S_t})) \tag{11}$$

$$\leq u(i_t(\boldsymbol{x}_{t,S_t})) - u(f(\boldsymbol{x}_{t,S_t})) \tag{12}$$

$$\leq B \sum_{k=1}^{|S_t|} |i_t(\overline{x}_{t,k}) - f(\overline{x}_{t,k})| \tag{13}$$

$$\leq B \sum_{k=1}^{|S_t|} \left|\mu_{[\![t-1]\!]}(\overline{x}_{t,k}) - f(\overline{x}_{t,k})\right|$$

$$+ B \sum_{k=1}^{|S_t|} \left|\beta_t^{1/2}\sigma_{[\![t-1]\!]}(\overline{x}_{t,k})\right| \tag{14}$$

$$\leq 2B\beta_t^{1/2} \sum_{k=1}^{|S_t|} \left|\sigma_{[\![t-1]\!]}(\overline{x}_{t,k})\right| , \tag{15}$$

where equation 10 follows from monotonicity of $u$ and the fact that $f(x_{t,s_{t,k}^*}) \leq i_t(x_{t,s_{t,k}^*})$, for $k \leq |S_t^*|$, by Lemma 2; equation 11 follows from the definition of $G_t$; equation 12 holds since $S_t$ is the super arm chosen by the $\alpha$-approximation Oracle; equation 13 follows from the Lipschitz continuity of $u$: equation 14 follows from the definition of index and the triangle inequality; for equation 15 we use Lemma 2. $\qquad \square$

We next provide the full proof of Lemma 1. We first restate the result below for convenience.

**Statement.** *Let $\boldsymbol{z}_t := \boldsymbol{x}_{t,S_t}$ be the vector of selected contexts at time $t \geq 1$. Given $T \geq 1$, we have:*

$$I\left(r(\boldsymbol{z}_{[T]}); f(\boldsymbol{z}_{[T]})\right) \geq \frac{1}{2(\sigma^{-2}\lambda^*(K) + 1)} \sum_{t=1}^{T} \sum_{k=1}^{|S_t|} \sigma^{-2}\sigma_{[\![t-1]\!]}^2(\overline{x}_{t,k}) ,$$

*where $\boldsymbol{z}_{[T]} = [\boldsymbol{z}_1, \ldots, \boldsymbol{z}_T]^T$ is the vector of all selected contexts until round $T$ and $\lambda^*(K)$ is the maximum eigenvalue of matrices $(\Sigma_{[\![t-1]\!]}(\boldsymbol{z}_{[t]}))_{t=1}^T$.*

*Proof.* By definition, we have

$$I\left(r(\boldsymbol{z}_{[T]}); f(\boldsymbol{z}_{[T]})\right) = H\left(r(\boldsymbol{z}_{[T]})\right) - H\left(r(\boldsymbol{z}_{[T]})\big|f(\boldsymbol{z}_{[T]})\right)$$

$$= H\left(r(\boldsymbol{z}_T), r(\boldsymbol{z}_{[T-1]})\right) - H\left(r(\boldsymbol{z}_{[T]})\big|f(\boldsymbol{z}_{[T]})\right)$$

$$= H\left(r(\boldsymbol{z}_T)\big|r(\boldsymbol{z}_{[T-1]})\right) + H\left(r(\boldsymbol{z}_{[T-1]})\right) - H\left(r(\boldsymbol{z}_{[T]})\big|f(\boldsymbol{z}_{[T]})\right) .$$

Reiterating inductively we obtain:

$$I\left(r(\boldsymbol{z}_{[T]}); f(\boldsymbol{z}_{[T]})\right) = \sum_{t=2}^{T} H\left(r(\boldsymbol{z}_t)\big|r(\boldsymbol{z}_{[t-1]})\right) + H\left(r(\boldsymbol{z}_1)\right) - H\left(r(\boldsymbol{z}_{[t]})\big|f(\boldsymbol{z}_{[t]})\right) ,$$

since $r(\boldsymbol{z}_{[1]}) = r(\boldsymbol{z}_1)$. Let $\boldsymbol{\eta}_t = [\eta_{t,1}, \ldots, \eta_{t,|S_t|}]^T$. Note that

$$
\begin{aligned}
H\left(r(\boldsymbol{z}_{[t]})\big|f(\boldsymbol{z}_{[t]})\right) &= H\left(f(\boldsymbol{z}_1) + \boldsymbol{\eta}_1, \ldots, f(\boldsymbol{z}_T) + \boldsymbol{\eta}_T\big|f(\boldsymbol{z}_{[T]})\right) \\
&= H\left(f(\overline{x}_{1,1}) + \eta_{1,1}, \ldots, f(\overline{x}_{1,|S_1|}) + \eta_{1,|S_1|}, \ldots, f(\overline{x}_{T,1}) \right. \\
&\qquad \left. +\eta_{T,1}, \ldots, f(\overline{x}_{T,|S_T|}) + \eta_{T,|S_T|}\big|f(\boldsymbol{z}_{[T]})\right) \\
&= \sum_{t=1}^{T}\sum_{k=1}^{|S_t|} H(\eta_{t,k}) \\
&= \frac{1}{2}\sum_{t=1}^{T} \log\left|2\pi e\sigma^2 \boldsymbol{I}_{|S_t|}\right| ,
\end{aligned}
$$

where the last equality follows from the entropy formula for Gaussian random variables. On the other hand, since any finite dimensional distributions associated with a Gaussian process are Gaussian random vectors and also, since $\boldsymbol{z}_t$ is deterministic given $\boldsymbol{z}_{[t-1]}$, the conditional distribution of $r(\boldsymbol{z}_t)$ given $r(\boldsymbol{z}_{[t-1]})$ is $\mathcal{N}\left(\mu_{[\![t-1]\!]}(\boldsymbol{z}_t), \Sigma_{[\![t-1]\!]}(\boldsymbol{z}_t) + \sigma^2 \boldsymbol{I}_{|S_t|}\right)$, where $\mu_{[\![t-1]\!]}(\boldsymbol{z}_t) = [\mu_{[\![t-1]\!]}(\overline{x}_{t,1}), \ldots, \mu_{[\![t-1]\!]}(\overline{x}_{t,|S_t|})]^T$ and

$$
\Sigma_{[\![t-1]\!]}(\boldsymbol{z}_t) = \begin{bmatrix} \sigma^2_{[\![t-1]\!]}(\overline{x}_{t,1}) & \cdots & k_{[\![t-1]\!]}(\overline{x}_{t,1}, \overline{x}_{t,|S_t|}) \\ \vdots & \ddots & \vdots \\ k_{[\![t-1]\!]}(\overline{x}_{t,|S_t|}, \overline{x}_{t,1}) & \cdots & \sigma^2_{[\![t-1]\!]}(\overline{x}_{t,|S_t|}) \end{bmatrix}.
$$

Here, $k_0(x,y) = k(x,y)$ and $k_{[\![t-1]\!]}(x,y) := k_N(x,y)$, where $N = \sum_{t'=1}^{t-1} |S_{t'}|$. Before we proceed, we need to emphasize the fact that O'CLOK-UCB does not employ any randomisation subroutines. There exist many deterministic $\alpha$-approximation oracles (Lin et al., 2015; Qin et al., 2014; Buchbinder & Feldman, 2018), including greedy approximation oracles. We assume that the $\alpha$-approximation oracle that our algorithm uses is deterministic, and hence, there is no randomness coming from our algorithm. Therefore, we obtain the following.

$$
\begin{aligned}
I(r(\boldsymbol{z}_{[T]}); f(\boldsymbol{z}_{[T]})) &= \sum_{t=2}^{T} H\left(r(\boldsymbol{z}_t\big|r(\boldsymbol{z}_{[t-1]}))\right) + H\left(r(\boldsymbol{z}_1)\right) - H\left(r(\boldsymbol{z}_{[t]})\big|f(\boldsymbol{z}_{[t]})\right) \\
&= \sum_{t=2}^{T} \frac{1}{2}\left[\log\left|2\pi e\left(\Sigma_{[\![t-1]\!]}(\boldsymbol{z}_t) + \sigma^2 \boldsymbol{I}_{|S_t|}\right)\right|\right] \\
&\quad + \frac{1}{2}\left[\log\left|2\pi e\left(\Sigma_0(\boldsymbol{z}_1) + \sigma^2 \boldsymbol{I}_{|S_1|}\right)\right|\right] - \frac{1}{2}\sum_{t=1}^{T} \log\left|2\pi e\sigma^2 \boldsymbol{I}_{|S_t|}\right| \quad\quad (16) \\
&= \sum_{t=1}^{T} \frac{1}{2}\log\left|2\pi e\left(\Sigma_{[\![t-1]\!]}(\boldsymbol{z}_t) + \sigma^2 \boldsymbol{I}_{|S_t|}\right)\right| - \frac{1}{2}\sum_{t=1}^{T} \log\left|2\pi e\sigma^2 \boldsymbol{I}_{|S_t|}\right| \\
&= \sum_{t=1}^{T} \frac{1}{2}\log\left|2\pi e\sigma^2\left(\sigma^{-2}\Sigma_{[\![t-1]\!]}(\boldsymbol{z}_t) + \boldsymbol{I}_{|S_t|}\right)\right| - \frac{1}{2}\sum_{t=1}^{T} \log\left|2\pi e\sigma^2 \boldsymbol{I}_{|S_t|}\right| \\
&= \sum_{t=1}^{T} \frac{1}{2}\log\left|2\pi e\sigma^2 \boldsymbol{I}_{|S_t|}\right| + \sum_{t=1}^{T} \frac{1}{2}\log\left|\left(\sigma^{-2}\Sigma_{[\![t-1]\!]}(\boldsymbol{z}_t) + \boldsymbol{I}_{|S_t|}\right)\right| \\
&\quad - \frac{1}{2}\sum_{t=1}^{T} \log\left|2\pi e\sigma^2 \boldsymbol{I}_{|S_t|}\right| \\
&= \frac{1}{2}\sum_{t=1}^{T} \log\left|\left(\sigma^{-2}\Sigma_{[\![t-1]\!]}(\boldsymbol{z}_t) + \boldsymbol{I}_{|S_t|}\right)\right| \quad\quad (17) \\
&= \frac{1}{2}\sum_{t=1}^{T} \log\left(\prod_{k=1}^{|S_t|}(\sigma^{-2}\lambda_k + 1)\right) \quad\quad (18)
\end{aligned}
$$

$$= \frac{1}{2} \sum_{t=1}^{T} \sum_{k=1}^{|S_t|} \log(\sigma^{-2} \lambda_k + 1)$$

$$\geq \frac{1}{2} \sum_{t=1}^{T} \sum_{k=1}^{|S_t|} \frac{\sigma^{-2} \lambda_k}{\sigma^{-2} \lambda_k + 1} \tag{19}$$

$$\geq \frac{1}{2(\sigma^{-2} \lambda^*(K) + 1)} \sum_{t=1}^{T} \sum_{k=1}^{|S_t|} \sigma^{-2} \lambda_k \tag{20}$$

$$= \frac{1}{2(\sigma^{-2} \lambda^*(K) + 1)} \sum_{t=1}^{T} \sum_{k=1}^{|S_t|} \sigma^{-2} \sigma^2_{[\![t-1]\!]}(\overline{x}_{t,k}) \tag{21}$$

where equation 16 follows from the formula of conditional entropy for Gaussian random vectors; for equation 18, we make the observation that given a symmetric positive-definite $n$ by $n$ matrix $A$, we can write $|A + \boldsymbol{I}_n| = \prod_{k \leq n} (\lambda_k + 1)$; for equation 19, we use the fact that $\log x \geq (x-1)/x$, for all $x > 0$; in equation 20 $\lambda^*(K)$ denotes the maximal eigenvalue of all $\Sigma_{[\![t-1]\!]}(\boldsymbol{z}_t)$, for $t \leq T$; for equation 21 we use the fact that the trace of a symmetric matrix is equal to the sum of its eigenvalues. $\qquad \square$

Finally, we are ready to prove Theorem 1.

**Statement.** *Let $\delta \in (0,1)$, $T \in \mathbb{N}$. Given $\beta_t = 2 \log(M \pi^2 t^2 / 3 \delta)$, the regret incurred by O'CLOK-UCB in $T$ rounds is upper bounded with probability at least $1 - \delta$ as follows:*

$$R_\alpha(T) \leq \sqrt{C \beta_T K T \gamma_{KT}(\mathcal{X}_{\overline{T}})} \ ,$$

*where $C = 8B^2(\lambda^*(K) + \sigma^2)$.*

*Proof.* From Lemma 3 we have:

$$R_\alpha(T) = \alpha \sum_{t=1}^{T} \mathrm{opt}(f_t) - \sum_{t=1}^{T} u(f(\boldsymbol{x}_{t,S_t}))$$

$$\leq 2B \beta_T^{1/2} \sum_{t=1}^{T} \sum_{k=1}^{|S_t|} \left| \sigma_{[\![t-1]\!]}(\overline{x}_{t,k}) \right| \ , \tag{22}$$

using the fact that $\beta_t^{1/2}$ is monotonically increasing in $t$. Taking the square of both sides, we have:

$$R_\alpha^2(T) \leq 4B^2 \beta_T \left( \sum_{t=1}^{T} \sum_{k=1}^{|S_t|} \left| \sigma_{[\![t-1]\!]}(\overline{x}_{t,k}) \right| \right)^2$$

$$\leq 4B^2 \beta_T T \sum_{t=1}^{T} \left( \sum_{k=1}^{|S_t|} \left| \sigma_{[\![t-1]\!]}(\overline{x}_{t,k}) \right| \right)^2 \tag{23}$$

$$\leq 4B^2 \beta_T T \sum_{t=1}^{T} |S_t| \sum_{k=1}^{|S_t|} \sigma^2_{[\![t-1]\!]}(\overline{x}_{t,k}) \tag{24}$$

$$\leq 4B^2 \beta_T T K \sum_{t=1}^{T} \sum_{k=1}^{|S_t|} \sigma^2_{[\![t-1]\!]}(\overline{x}_{t,k})$$

$$= 4B^2 \beta_T T K \sigma^2 \sum_{t=1}^{T} \sum_{k=1}^{|S_t|} \sigma^{-2} \sigma^2_{[\![t-1]\!]}(\overline{x}_{t,k}) \tag{25}$$

$$\leq 8B^2\beta_T TK(\sigma^{-2}\lambda^*(K)+1)\sigma^2 I\left(r(\boldsymbol{z}_{[T]}); f(\boldsymbol{z}_{[T]})\right) \tag{26}$$

$$\leq CK\beta_T T\gamma_{KT}(\mathcal{X}_{\overline{T}}) \, , \tag{27}$$

where for equation 23 and equation 24, we have used the Cauchy–Schwarz inequality twice; in equation 25, we just multiply by $\sigma^2$ and $\sigma^{-2}$; equation 26 follows from Lemma 1; and for equation 27, we use the definition of $\overline{\gamma}_T$.

Taking the square root of both sides we obtain our desired result. $\square$

Finally, we prove Corollary 1.

**Statement.** *Let $\delta \in (0,1)$, $T,K \in \mathbb{N}$ and let $\mathcal{X} \subset \mathbb{R}^D$ be compact and convex. Under the conditions of Theorem 1 and for the following kernels, the $\alpha$-regret incurred by O'CLOK-UCB in $T$ rounds is upper bounded (up to polylog factors) with probability at least $1 - \delta$ as follows:*

- *For the linear kernel we have: $R_\alpha(T) \leq \tilde{O}\left(\sqrt{\lambda^*(K)DKT}\right)$.*

- *For the RBF kernel we have: $R_\alpha(T) \leq \tilde{O}\left(\sqrt{\lambda^*(K)KT\log^D T}\right)$.*

- *For the Matérn kernel we have: $R_\alpha(T) \leq \tilde{O}\left(\sqrt{\lambda^*(K)KT^{(D+\nu)/(D+2\nu)}}\right)$, where $\nu > 1$ is the Matérn parameter.*

*Proof.* This is an immediate application of the explicit bounds on $\gamma_T$ given in Theorem 5 of (Srinivas et al., 2012), to the bound we obtained in Theorem 1. For the Matérn kernel, we have used the tighter bounds from (Vakili et al., 2020). $\square$

