# OpenReview forum: "Contextual Combinatorial Bandits With Changing Action Sets Via Gaussian Processes"
_TMLR — Accepted by TMLR_

### Review · Reviewer_yA9y · 2025-06-09

**Summary Of Contributions:**

- This paper formulates combinatorial contextual multi-armed bandits with changing action sets (C3 MAB), where the underlying outcomes of base arms are sampled from a Gaussian process.
- This paper proposes a UCB-based acquisition function named O'CLOCK-UCB, which achieves a sub-linear (regarding the number of iterations $T$) high-probability regret upper bound.
- To obtain the regret upper bound, the authors show the lower bound of the maximum information gain (Lemma 3).

**Audience:**

Yes

**Claims And Evidence:**

No

**Requested Changes:**

Please address the above weaknesses and careless mistakes. This revision should include the modification (or removal) in Section 1 related to those. Furthermore, if the authors agree, please consider the revision addressing minor comments.

**Strengths And Weaknesses:**

**Strength**:
- To my knowledge, Lemma 3 seems to be novel and is interesting.

**Weakness**:
- In contributions, the authors claim that $\bar{\gamma}\_T$ takes into account the availability of time-varying action sets. However, if the contexts of base arms can take arbitrary elements in $\mathcal{X}$, then this quantity $\bar{\gamma}\_T$ has the same rate as $\gamma\_{KT}$. Indeed, the resulting explicit upper bound in Corollary 2 is given by using $\gamma\_{KT}$. Therefore, I think that if the authors want to claim the benefit from using the quantity $\bar{\gamma}\_T$, the authors should provide more concrete analysis which shows $\bar{\gamma}\_T$ has a rate strictly slower than $\gamma\_{KT}$ under some conditions of super arms.
- In Section 1.2, the authors describe that ``while they take a Bayesian approach to tackling it, our approach is frequentist.'' However, in the GP-bandit literature, the frequentist setting implies that the setting where $f$ belongs to the reproducing kernel Hilbert space. On the other hand, this paper assumes that $f$ follows a Gaussian process, which is often called the Bayesian setting. I believe that the proposed method in this paper is a Bayesian approach. The difference from Sandberg et al., 2023 is whether the target is the high probability regret bound or the expected regret.
- Lemma 3 seems to be novel. However, the maximal eigenvalue $\lambda^* (K)$ can be $K$ in the worst case, as discussed in Remark 1. If we substitute $K$ for $\lambda\^*(K)$, a similar result to Lemma 3 can be obtained by Proposition 1 in Desautels et al., 2014 (Consider the case where the all observations in $y_{fb[t]+1:t-1}$ is on $x$). Indeed, the batched Bayesian optimization problem is similar to C3 MAB in this paper in the sense that users must choose action sets before observations are obtained. Although I think Lemma 3 has a certain value, a more careful discussion of the relationship between this paper's proof and the proof technique in batched Bayesian optimization literature is required.

>Thomas Desautels, Andreas Krause, Joel W. Burdick; 15(119):4053−4103, 2014. Parallelizing Exploration-Exploitation Tradeoffs in Gaussian Process Bandit Optimization

- Regarding Remark 1, if the covariance matrix is always diagonal, $\gamma_T = \Omega(T)$. Therefore, this discussion is misleading and should be removed.
- The upper bound in Theorem 2 is obvious since the base arms in $\mathcal{X}$. Furthermore, the lower bound seems to be false. The derivation in Eq. (31) does not hold. This is because the maximization is performed on $\mathcal{X}$ in Eq. (31) but on $\cup_t \mathcal{X}_t$ in Eq. (30), and $\mathcal{X}_t$ can be arbitrarily small compared with $\mathcal{X}$. I think that Theorem 2 should be removed or changed to a lemma, a proposition, and so on.
- To me, the derivation of Corollary 2 is unclear. For example, why does the upper bound in the RBF kernel contain $\sqrt{D}$ term? Why does the upper bound in the Matern kernel contain $\lambda^* (K)$ term (not $\sqrt{\lambda^* (K)}$)? Why does the upper bound in the Matern kernel not contain $K$?
- Although authors claim that the computational complexity of O'CLOCK-UCB is $O(K^3 T^4)$. However, if we update the inverse kernel matrix using the inversion of the block submatrices, the resulting computation complexity is $O(K^3 T^3)$. Therefore, the experimental result of O'CLOCK-UCB on the computational time may be slow due to the implementation.


**(Probably) Careless mistakes**:
- Takeno et al., 2023ab did not perform the analyses based on adaptive discretization.
- In the fourth contributions listed in page 2, $K$ in $\gamma\_{KT}$ is missing.
- In Remark 1, is the word "not" in the statement "If that was the case" missed? Please rephrase for clarification.
- In Corollary 1, the last phrase "with cardinality $KT$" seems to be a mistake of $\bar{T}$.
- The upper bounds in Corollary 2 in the main paper are different from the statement in the appendix.
- In the experiments, the abbreviation DPMC is not defined.

**Minor comments**:
- The whole paper is not easy to read for me. In particular, Section 2 is unclear since the problem setup, assumptions, and example applications are discussed in an inconsistent order (The order seems to be Definition of $f$ -> definition of $u$ -> assumptions on $u$ -> Definition of bandit problem -> Example applications -> Assumption on $f$). Furthermore, Section 5 seems to contain redundant content. In addition, I cannot see the necessity of several slightly complex notations, such as tilde and bar on $x$ and $z$.
- The description "the expected reward is Lipschitz continuous in expected base arm outcomes" may be confusing. Since we consider that $f$ is a sample path from a GP, the expected base arm outcomes can also be interpreted as $\mu\_{t-1}(x)$, not $f(x)$.
- Regarding the description "We assume that $\eta$ is independent across base arms," the authors assume that $\eta$ is mutually independent across all observations, not only across base arms.
- Remark 2 is slightly unclear since Lemma 3 is not explained in detail in the main paper. Moreover, since a similar proof technique has been considered in the batched Bayesian optimization literature, "a new approach" may be overemphasized.
- I do not think that Remark 3 is remarkable.

---

> ### Author Response · Authors · 2025-07-11
> **Rebuttal**
>
> We thank the reviewer for the very insightful and helpful comments! We address them below.
>
> **Comment:** In contributions, the authors claim that \gamma_\bar{T} takes into account...
>
> **Response:**  We agree with the reviewers that, in general, we can only show $\overline{\gamma}\_{T}\leq \gamma\_{KT}$. There are, nonetheless, examples where such an inequality is strict. Since we do not have access to the arm availability model over time, we cannot be sure that some arms may never appear during $T$ rounds. Consider the following example. Let the sequence $\mathcal{X}\_t$ up to round $T$ be disjoint and let their union be a strict subset of $\mathcal{X}$. Furthermore, assume that the elements not contained in this union are more than $K$, and denote by $\tilde{x}^\star\_K:=(x_1^\star,\ldots,x^\star\_K)$ the contexts not in the union that satisfy $\tilde{x}^\star\_K = \arg\max\_{x\_K \in \mathcal{X}^K} I(r(x\_K), f(x\_K)).$
> Assume that such maximum is uniquely achieved by this subset, and that every other subset of $\mathcal{X}$ achieves an information gain which is strictly smaller than the one achieved by $\tilde{x}^\star\_K$. As a consequence, it is easy to see that $\bar{\gamma}\_T < \gamma\_{KT}$, since, in the time-varying actions model, the unique information-maximizing super-arm is never seen.
>
> **Comment:** In Section 1.2, the authors describe that ``while they take a Bayesian...
>
> **Response:** We thank the reviewer for the clarification they provided. We have changed the discussion in Section 1.2 accordingly (all changes appear in blue). Also, we agree with the reviewer that the difference from Sandberg et al. (2023) is that they consider the expected (Bayesian cumulative) regret, while we are interested in minimizing the $\alpha$-approximation regret. We have included such a difference in the related work section.
>
> **Comment:** Lemma 3 seems to be novel. However, the maximal eigenvalue...
>
> **Response:** We thank the reviewer for pointing us to additional literature. While we see that there may be a certain similarity in flavor of Lemma 3 to Proposition 1 of the referenced paper, we believe they point to different properties. Proposition 1 provides a measure of proportionality between variances using the information gain. While this property can be used in favor of our result (although it is not clear if it directly follows solely from it), it does not capture the extent of Lemma 3, or its purpose. In its proof, we have also used different matrix analysis techniques and properties of the entropy which are not needed for the purposes of Proposition1.
>
> At this point, it might also be useful to emphasize the difference between the batched Bayesian optimization setting and the C3MAB setting. Note that, in both cases, batches of arms (or superarms) have to be selected previous to observing final feedback. This, together with the Bayesian modelling (in our case, a GP instance), are common characteristics of both settings. However, we further extend the classical formulation of the CMAB to (i) allowing contextual information for each arm, (ii) allowing for infinitely many arms in our arm set and, crucially, (iii) allowing for volatile arm sets. While these additional properties can be incorporated in the batched Bayesian setting, in this paper, we take the approach of studying them in the C3MAB context.
>
> **Comment:** Regarding Remark 1, if the covariance matrix is always diagonal...
>
> **Response:** We respectfully disagree with the reviewer that Remark 1 is misleading. While $\gamma_T=\Omega(T)$ may be the case for diagonal covariance matrices, this does not contradict the $O(\sqrt{K\bar{\gamma}_T})$ bounds, since $K$ might be a factor inside the information gain term in the general case.
>
> **Comment:** The upper bound in Theorem 2 is obvious since the base arms in \cal{X}...
>
> **Response:** We would kindly like to take the reviewer’s attention to the statement of Theorem 1. We specifically provide a condition under which the lower bound holds, namely, that volatility is instantiated as the classical setting where all arms are available. In that case, the maximization over the union of available contexts in every round is identical to the maximization over the whole context space.
>
> **Comment:** To me, the derivation of Corollary 2 is unclear. For example, why does the upper bound...
>
> **Response:** We thank the reviewer for pointing out the typos in the statements. We have corrected the statements in the current version of the paper.

---

> ### Author Response · Authors · 2025-07-11
> **Rebuttal**
>
> **Comment:** Although authors claim that the computational complexity of O'CLOCK-UCB...
>
> **Response:** We sincerely thank the reviewer for this comment and for identifying this crucial detail in our complexity analysis. The reviewer is absolutely correct. With an efficient implementation using sequential updates via block matrix inversion, the total computational complexity for the non-sparse O'CLOK-UCB is indeed $O(K^3T^3)$, not the $O(K^3T^4)$ implied by a naive re-computation. We apologize for this oversight and have corrected the analysis in the manuscript. Our experiments utilize the GPflow library, which already employs highly optimized routines consistent with this efficient approach. The steep computational curve seen in Figure 4 reflects the polynomial growth of the per-round cost, which is expected and aligns with the overall $O(K^3T^3)$ complexity.
>
> **Comment:** Careless Mistakes (all of them)
>
> **Response:** We have addressed all mentioned comments in the current version of the paper.
>
> **Comment:** The whole paper is not easy to read for me. In particular, Section 2 is unclear ...
>
> **Response:** We thank the reviewer for the feedback on readability. We have carefully re-read Section 2 and polished the text to improve the flow. We believe the current structure is logical, but we will ensure notation is as clear as possible in the final version. We have also removed repetitions in Section 5 and made it more concise. Note that the ordering in Section 2 does not contain the definition of a bandit problem before examples. The bandit problem is in itself an example of the C3-MAB framework which we introduce. Thus it is seen as part of the examples subsection. We have also removed the unnecessary tildes in Section 2 from notation.
>
> **Comment:** The description "the expected reward is Lipschitz continuous in expected base ...
>
> **Response:** We would like to point out to the reviewer that $\boldsymbol{f}$ denotes the expected arm outcomes, while $\mu$ is reserved for the posterior mean of the GP. These two notions are not the same thing. That is why we use different notations.
>
> **Comment:** Regarding the description "We assume that eta is independent across base arms,"...
>
> **Response:** We thank the reviewer for the clarification. We have rephrased the assumption in the current version of the paper.
>
> **Comment:**  Remark 2 is slightly unclear since Lemma 3 is not explained in detail in the...
>
> **Response:** Please refer to the earlier discussion on Lemma 3. Based on it, we do believe that Lemma 3 does indeed use a new approach. However, we have omitted the phrase as per the reviewer’s suggestion.
>
> **Comment:** I do not think that Remark 3 is remarkable.
>
> **Response:** We have kept the text as normal text.

---

> > ### Comment · Reviewer_yA9y · 2025-07-17
> >
> > I appreciate the detailed response and clarification. Thanks to the clarifications, I noticed what I misunderstood. However, I still have several concerns listed below:
> >
> > ## Concerns
> >
> > >Comment: In contributions, the authors claim that \gamma\_\bar{T} takes into account...
> >
> > I still have the following concerns about using a quantity $\overline{\gamma}\_T$.
> >
> > 1. One reason I suggest rephrasing $\overline{\gamma}\_T$ is that $\overline{\gamma}\_T$ is the quantity whose rate is difficult to comprehend at first glance for GP bandit researchers. Furthermore, to analyze the example that the authors showed, the quantity $\gamma\_{KT}(\cup\_i \mathcal{X}\_i)$ (or $\gamma\_{\overline{T}}(\cup\_i \mathcal{X}\_i)$), where I define $\gamma\_T (\mathcal{X})$ as the maximum information gain of $T$ observations in $\mathcal{X}$, seems to be sufficient. In addition, this quantity is the usual maximum information gain which is easy to understand for GP bandits researchers.
> > 2. I believe that the upper bound should be shown by quantities whose rates are explicitly known. In that sense, if the authors finally show the explicit rate through $\gamma\_{KT}$, it is not very meaningful to show the upper bound by $\overline{\gamma}\_T$. For example, it is well-known that $\sum\_{t=1}^T \sigma^2\_t (x\_t) \leq C\_1 \gamma\_T$ with some $C\_1 > 0$ and in most cases, this inequality is strict. However, most studies show the regret upper bound by $\gamma\_T$ since a tighter upper bound of $\sum\_{t=1}^T \sigma^2\_t (x\_t)$ than $\gamma\_T$ is not known, and the explicit rate of $\gamma\_T$ is well-investigated.
> >
> > Consequently, I still consider that, if the authors cannot provide a more concrete benefit of using $\overline{\gamma}\_T$, it is better to rephrase the upper bound by $\gamma\_{KT}(\cup\_i \mathcal{X}\_i)$, which is easy to understand.
> >
> > >Comment: Regarding Remark 1, if the covariance matrix is always diagonal...
> >
> > From the following reasons, I consider that the discussion in Remark 1 is unrealistic.
> >
> > 1. If we apply the linear, RBF, or Matern kernel to $\mathcal{X}$ naively, all input $x \in \mathcal{X}$ have correlations with each other in the prior. Therefore, except for an extreme case, the independence of the arms does not hold.
> > 2. Even if there are at least $|S\_t|$ independent arms, the proposed method does not have a mechanism to select independent candidates.
> >
> > I believe that the justification based on unrealistic assumptions is misleading.
> > Therefore, I consider that the authors should remove Remark 1 or provide a more concrete example in which the independence assumption holds to make Remark 1 meaningful.
> >
> >
> > On the other hand, if the proposed method chooses independent arms in all iterations, then the maximum information gain can be $O(K \gamma\_T)$ due to independence. Therefore, although $\lambda^*(K) \leq 1$, the resulting regret upper bound can be $\tilde{O}(K \sqrt{T \gamma\_T})$, which is linear in $K$.
> > Thus, even if the independence assumption holds, the claim of Remark 1 appears to be still misleading since this dependence on $K$ in $\overline{\gamma}\_T$ is not discussed explicitly.
> >
> >
> > >Comment: The upper bound in Theorem 2 is obvious since the base arms in \cal{X}...
> >
> > I am sorry that I missed the conditions and thank the clarification.
> > However, if this condition $|S\_t| = K$ and $\mathcal{X}\_t = \mathcal{X}$ holds for all $t$, then $\overline{\gamma}\_T = \gamma\_{KT}$ seems to hold.
> > Is this correct?

---

> > > ### Comment · Reviewer_yA9y · 2025-07-17
> > >
> > > ## Minor comments
> > >
> > > >We provide a connection between $\overline{\gamma}\_T$ and the classical maximum information gain (Srinivas et al., 2012), denoted by $\gamma\_T$, and characterize the role that $K$ plays in these bounds (see Theorem 3).
> > >
> > > The theorem number is not consistent with the current version of the paper.
> > >
> > > >The main difference between a frequentist approach, such as adaptive discretization, and a Bayesian one, such as GPs, is that, while adaptive discretization discretizes the search space into regions and maintains statistics over each region, GPs offer a functional approach.
> > >
> > > In GP bandit literature, a frequentist approach often means that the GP posteriors are used to construct the (non-Bayesian) confidence interval to predict a black box function that belongs to the reproducing kernel Hilbert space (RKHS) endowed by a known kernel function, e.g., see Theorem 3 of Srinivas et al., 2010, and Chowdhury and Gopalan, 2017. In this case, the frequentist approach does not discretize the search space into regions, but rather uses the posterior mean and variance to predict the function value at any point in the search space. Therefore, I suggest rephrasing the paragraph starting with this sentence to avoid confusion. (I feel that this paragraph states the difference from the adaptive discretization-based approaches, not the difference from the frequentist approach.)
> > >
> > >
> > > 1. Niranjan Srinivas, Andreas Krause, Sham M. Kakade, Matthias Seeger, Gaussian Process Optimization in the Bandit Setting: No Regret and Experimental Design, 2010.
> > > 2. Sayak Ray Chowdhury, Aditya Gopalan, On Kernelized Multi-armed Bandits, 2017.
> > >
> > >
> > > >Moreover, while they consider the notion of expected (Bayesian cumulative) regret, we are
> > > interested in minimizing the notion of α-approximation regret.
> > >
> > > I think that it should be stated that the high-probability regret bound is derived to make the difference from Sandberg et al. (2023) clearer.
> > >
> > >
> > >
> > > >Comment: To me, the derivation of Corollary 2 is unclear. For example, why does the upper bound...
> > >
> > >  I thank the clarification. However, if the authors write the dependence of $\sqrt{D}$ for the linear kernel, then it appears natural to write $\log^{D/2} T$ for the RBF kernel as well.

---

> ### Author Response · Authors · 2025-07-18
> **Rebuttal**
>
> We thank the reviewer for the follow-up comments. We address them below.
>
> **Comment:** I still have the following concerns about using a quantity \bar\gamma_T...
>
> **Response:** We have changed the definition of the maximum information gain (Equation (4)) into the one the reviewer suggests. The current manuscript reflects all changes in blue.
>
> **Comment:** From the following reasons, I consider that the discussion in Remark 1 is unrealistic...
>
> **Response:** We have removed Remark 1 from the current version of the manuscript.
>
> **Comment:**  I am sorry that I missed the conditions and thank the clarification...
>
> **Response:** Yes, the reviewer is right that, in that case, we are left with equality. Using the classical notion (adapted to our case) which the reviewer has suggested resolves all problems related to this as we now have also removed the previous result comparing the different notions.
>
> **Comment:** The theorem number is not consistent with the current version of the paper.
>
> **Response:** We have made the numbering consistent.
>
> **Comment:** In GP bandit literature, a frequentist approach often means that the GP posteriors...
>
> **Response:** We thank the reviewer for the detailed insights into the frequentist approach! We have updated the paragraph (in the Related Work section) accordingly.
>
> **Comment:** I think that it should be stated that the high-probability regret bound is derived...
>
> **Response:** We thank the reviewer for catching this detail. We have updated the phrasing accordingly.
>
> **Comment:**  I thank the clarification. However, if the authors write the dependence of  for the linear kernel...
>
> **Response:** We agree with the reviewer and have updated the manuscript accordingly (with changes reflected in Table 1, Corollary 1 and its restatement in the Appendix).

---

> > ### Comment · Reviewer_yA9y · 2025-07-22
> >
> > I appreciate the revisions. My concerns are almost resolved, but the following seems not revised:
> > - In the third item of Sec. 1.1, the description about the diagonal covariance matrices is not removed.
> > - Regarding Corollary 1, although I might have written an ambiguous description, the known upper bound of maximum information gain associated with the RBF kernel is $O(\log^{d + 1} T)$. Therefore, in Corollary 1 and the appendix, I think that the regret upper bound for the RBF kernel should be $O(\sqrt{\lambda^*(K) KT \log^D T})$ (if we ignore the dimension-independent $\log T$ term).
> > - In Sec. 6, the regret upper bound is not revised.
> >
> > Finally, I would like to recommend that the authors carefully recheck the entire paper, as I may have also overlooked issues like the above.

---

> > > ### Author Response · Authors · 2025-07-23
> > > **Response to Official Comment by Reviewer yA9y**
> > >
> > > We thank the reviewer for their detailed reviewing of our paper, and we sincerely apologize for any details that we overlooked! We have made sure to incorporate all the typos/issues that the reviewer noticed and have revised the paper for any additional details that were overlooked. All relevant changes are reflected in the current version of the manuscript.

---

### Review · Reviewer_WMHd · 2025-06-26

**Summary Of Contributions:**

This paper studies the contextual combinatorial bandit problem, where the available action sets can vary across rounds. Solving such problems without any structural assumptions would be extremely challenging and, in many cases, impractical. To address this, the authors introduce a smoothness assumption using Gaussian Processes (GPs), which can flexibly model a wide range of function classes. The resulting regret bound incorporates a notion of maximum information gain, but instead of relying on the worst-case information gain over all possible (and potentially unreachable) action sets, it focuses on the maximum achievable information gain within the action sets that can actually appear during the learning process. This would lead to a more meaningful and potentially tighter regret characterization.

**Audience:**

Yes

**Claims And Evidence:**

Yes

**Requested Changes:**

1. Regarding the above weakness, it would be helpful to further elaborate on how introducing GPs differs from relying on adaptive discretization or structural assumptions as in previous work. While using GPs removes the need for adaptive discretization (which is known to be efficient, as the authors noted on p.3), it introduces significant computational overhead. Additionally, although SO'CLOK-UCB is more efficient than O'CLOK-UCB, it still incurs considerably higher computational cost compared to CC-MAB. Providing a more detailed explanation of the practical and theoretical benefits of using GPs, beyond the improved regret bound, would strengthen the case for the proposed approach (if possible).

2. In Remark 1, the notation should be $\tilde{O}$ rather than $O$, due to the logarithmic dependency on $T$ through $\beta_T$.

3. In Figure 4, the results for ACC-UCB are not clearly visible (possibly due to overlap with other lines). It would be helpful to use a different line style or marker to distinguish the ACC-UCB results, or to explicitly discuss their behavior in the main text to ensure clarity.

---
### Very minor points (e.g. typos)

* In the definition of $\mathbf{r}\_{[N]}$ on p.7, I think $r(\tilde{x})$ instead of $r\_{[N]}(\tilde{x})$.
* There is a missing closing parenthesis ")" in the definition of $K_{[N]}$ right after equation 3.
* In Remark 2, it would be clearer to explicitly mention that Lemma 3 appears "in the appendix."
* In Lemma 3, the notation $\Sigma$ is introduced without definitions. Although it is a conventional notation, it would be helpful to explicitly clarify that it refers to the covariance matrix for completeness.

**Strengths And Weaknesses:**

### Strength

* Overall, the paper is well-written and easy to follow, with multiple illustrative examples of the C3-MAB setting and clear explanations of the proposed algorithm.
* The introduction of the maximum information gain over the sequence of observed contexts (of length $T$) is a reasonable and intuitive refinement. Considering the maximum over all possible contexts would be too conservative, especially in problems with large or infinite context spaces.
* The regret bound improves upon previous results when using commonly applied kernels, such as the linear and RBF kernels, making the contribution practically relevant.

### Weakness

The current manuscript may, at first glance, appear to be a straightforward adaptation of GP techniques. While the improved regret bounds are appreciated, the theoretical contribution seems somewhat incremental, as the core proof strategy largely follows established GP bandit analysis frameworks.

The primary theoretical novelty seems to lie in Lemma 3, which characterizes the maximum information gain over the observed context sequences. However, this result is established for a deterministic version of the proposed algorithm, which is computationally expensive and not scalable. Although the authors introduce a sparse variant to improve efficiency, this version incorporates additional randomness due to the selection of inducing points, which could impact the theoretical guarantees but is not fully analyzed in the paper.

---

> ### Author Response · Authors · 2025-07-11
> **Rebuttal**
>
> We thank the reviewer for the very insightful and helpful comments! We address them below.
>
> **Comment:** The current manuscript may, at first glance, appear to be a straightforward adaptation...
>
> **Response:** While the regret analysis follows a standard structure, we would like to emphasize that key results which capture the intricacies of our setting, such as Lemma 3 and Theorem 2, are completely novel. They in fact allow us to derive the desired upper bounds in this setting and compare the new notion of information gain with its classical counterpart. We have utilized a specific decomposition of the information gain, and have subsequently used various properties of entropy for Gaussian variables, together with algebraic manipulations to arrive at the desired result.
>
> **Comment:** The primary theoretical novelty seems to lie in Lemma 3, which characterizes...
>
> **Response:** We thank the reviewer for this important point. As the reviewer correctly notes, our theoretical guarantees are established for the non-sparse O'CLOK-UCB. The sparse variant is indeed introduced to address the practical challenge of computational efficiency.
>
> Our experiments (e.g., Figure 3) are designed to clearly demonstrate the expected trade-off: as the number of inducing points increases, the performance of the sparse algorithm closely converges to that of the original. While a full theoretical analysis of the sparse variant, which must account for the additional randomness of inducing point selection, presents significant technical challenges beyond the scope of this paper, we believe our comprehensive empirical results provide strong evidence for its reliability. They show that the sparse approach effectively balances high performance with computational tractability, making it a viable and powerful tool for practical applications.
>
> **Comment:** Regarding the above weakness, it would be helpful to further elaborate...
>
> **Response:** The main difference between a frequentist approach, such as adaptive discretization, and a Bayesian one, such as GPs, is that, while adaptive discretization discretizes the search space into regions and maintains statistics over each region, GPs offer a functional approach. Such a functional approach to maintaining valuable information allows us to retrieve point-wise statistical information in a fine-grained way just by evoking the posterior mean and variance. On the other hand, if one needs the same information in the adaptive discretization scenario, one would need to be satisfied with using the same quantities for all points in the same region. Consequently, adaptive discretization uses less computational resources, since it focuses only on ‘relevant’ regions and adaptively refines them further and further, while sacrificing a little bit of resolution. In contrast, GPs offer a high-resolution picture of all relevant statistics over the search space, and thus allow for a more precise inference, with an additional computational cost. We have resolved this latter issue by proposing a practical version of our algorithm. Furthermore, we have provided an experimental comparison between GPs and adaptive discretization, thus showcasing the benefit of using GPs as opposed to the latter.
>
> We hope we have provided a detailed comparison between both methods and clarified the benefit of using our method. We have added this discussion in the current version of the paper. (All changes appear in blue).
>
> **Comment:** In Remark 1, the notation should be...
>
> **Response:** We thank the reviewer for pointing out the typo. We have fixed it accordingly.
>
> **Comment:** In Figure 4, the results for ACC-UCB are not clearly visible...
>
> **Response:** We thank the reviewer for this observation. The line for ACC-UCB is indeed obscured by CC-MAB's in Figure 4. This is because both non-GP algorithms have very low and similar runtimes, causing their lines to overlap near the bottom of the plot. As suggested, we have clarified this directly in the main text to ensure the reader understands their relative performance.
>
> **Comment:** Very Minor Points (all of them)
>
> **Response:**  We thank the reviewer for pointing out these points! We have integrated all of the changes into the current version of the paper.

---

> > ### Comment · Reviewer_WMHd · 2025-07-22
> >
> > Thank you for the detailed response and clarification.
> > Most of my concerns are resolved. I just have one follow-up question regarding the following sentence:
> > > On the other hand, if one needs the same information in the adaptive discretization scenario, one would need to be satisfied with using the same quantities for all points in the same region.
> >
> > Could you clarify what is meant by "quantities" in this context?

---

> > > ### Author Response · Authors · 2025-07-23
> > > **Response to Official Comment by Reviewer WMHd**
> > >
> > > We thank the reviewer for the follow-up question. What we actually mean is the statistical information regarding the points in the region. We have updated the paragraph to reflect the answer to this question. The modified relevant sentence is reproduced below for convenience:
> > >
> > > "On the other hand, adaptive discretization operates on a lower resolution, maintaining the same statistical information for all points in the same region."

---

### Review · Reviewer_7ZVz · 2025-06-29

**Summary Of Contributions:**

The paper studies the problem of combinatorial semi-bandit in the case where the set of arms to be chosen varies over time. The authors provide an algorithm showing regret bounds and favourable empirical properties over synthetically generated data.

**Audience:**

Yes

**Broader Impact Concerns:**

There is not broad impact statement, but since the problem analysed is mainly theoretical I do not think it would be relevant to add one.

**Claims And Evidence:**

Yes

**Requested Changes:**

- \forall M \leq |S| -> \forall m \in \{1, \ldots |S|\}
-Example on page 6 about MAB: this is an example of the setting that can be modeled as a C3MAB, but it does not provide a practical reason to include in the modeling a time-varying set of arms. I think you should provide stronger examples for motivating your setting. Please provide a more convincing motivation.
-Examples about Online Shortest Path and Online stochastic ranking: not clear why you are presenting these examples
- The algorithm in Algorithm 1, even if for a specific advertising problem, has already been presented in the literature. You should refer properly to the corresponding literature:
Nuara, Alessandro, et al. "Online joint bid/daily budget optimization of internet advertising campaigns." Artificial Intelligence 305 (2022): 103663. or Accabi et al. "When Gaussian Processes Meet Combinatorial Bandits: GCB".
- I do not understand the rationale behind the transforming function in Section 5.1.1
- Simulation I: It is not clear why you tested your algorithm in an environment which is not reflect the assumptions you posed in the theoretical part of the paper. I would either select a problem which is more suitable to your setting, or also analyse the case in which we have an (\alpha, \beta)-approximate oracle.
-Simulation I: Also, the choice of experimenting with only the sparse version of the algorithm is not motivated. I would also like to understand how the original algorithm is behaving empirically.
- you should give some reference for the TIM+ oracle
- "Note that the reason ..." I think this comment was left from a previous version of the paper. I would suggest polishing and checking for such errors in the entire paper.
- The parameter setting in the second simulation for so'clok.-ucb is a repetition of the first one. Please rearrange so that you do not have to make so many repetitions in the paper.
- Figures: it is hard to read the legend...the lines are too thin. Please consider improving the readability of the graphs
- Figure 6: why not to compare the algorithms over the same environment? The choice of Figure 6 does not allow for comparison of the different algorithms.

**Strengths And Weaknesses:**

The paper is well-written, but the writing can be improved, as well as the structure in some of its parts.
I think that the authors overlooked some literature that proposed the same algorithm they are providing in this paper. Therefore, the main novelty lies in the theoretical derivation and its extension to varying arms.

I am not totally convinced by the motivating examples. The authors show some problems, but fail to make the reader understand how they correspond to the provided setting.

Finally, the experiment. I think they are somehow convincing; however, I think that they should be presented in a different way, first comparing the regular version of the algorithm they proposed with the sparse version, and then going more in-depth while discussing the results.

---

> ### Author Response · Authors · 2025-07-11
> **Rebuttal**
>
> We thank the reviewer for the very insightful and helpful comments! We address them below.
>
> **Comment:** I am not totally convinced by the motivating examples...
>
> **Response:** We have removed some of the examples that may be considered as instances of the broader CMAB setting instead of capturing the time-varying arm set feature of our setting. However, we believe that the remaining examples faithfully capture the intricacies of our setting and represent instances of it. We have explained how both examples have time-varying arm sets and what the outcomes and the final combinatorial objectives are for both examples. If the reviewer has a more specific question regarding some of the examples, we would happily try to address it.
>
> **Comment:** \forall M \leq |S| -> \forall m \in {1, \ldots |S|} -Example on page 6 about MAB...
>
> **Response:** All provided examples are instances of the C3MAB framework. However, the first instance might be too trivial, while we agree with the reviewer that there is no special need to include the last two examples as instances of C3MAB since they are special cases. Therefore, we have removed the first and last two examples from the list.
>
> **Comment:** The algorithm in Algorithm 1, even if for a specific advertising problem, has already been presented...
>
> **Response:** We thank the reviewer for pointing out an additional reference. We have included the following discussion in the current version of the paper.
>
> “Additionally, (Accabi et al., 2018) and (Nuara et al., 2022) propose algorithms with a similar
> rationale to O’CLOK-UCB. However, while our algorithm is provably no-regret in the C3MAB setting with
> (potentially) infinitely many context-dependent time-varying arms, the setting considered in (Accabi et al.,
> 2018) and (Nuara et al., 2022) is a standard combinatorial semi-bandit setting with a context-free time-
> invariant finite arm set.”
>
> **Comment:** I do not understand the rationale behind the transforming function...
>
> **Response:** We chose this scaled logistic (sigmoid) function, $f(x_{i,j}) = 2/(1+e^{-4x_{i,j}}) - 1$, to model a realistic non-linear relationship. Its S-shape captures a natural saturation effect, where the outcome is most sensitive to mid-range context values but has diminishing returns at the extremes. This creates a challenging but realistic expected outcome function.
>
> **Comment:** Simulation I: It is not clear why you tested your algorithm in an environment which...
>
> **Response:** We designed Simulations II and III to directly validate our theory, as they perfectly fit the paper's assumptions, including the use of an $\alpha$-oracle. Simulation I, on the other hand, was included to evaluate the practical robustness of O’CLOK-UCB in a more challenging and realistic setting. This simulation reflects a more challenging and realistic environment where the underlying optimization problem is NP-hard and the standard solution involves an $(\alpha, \beta)$-approximate oracle. While our theory requires an $\alpha$-oracle, the strong performance in Simulation I shows that our algorithm is effective and has no issues working with this more general class of oracles in practice, highlighting its broad applicability.
>
> Regarding the use of only the sparse version in Simulation I, this decision was based on both computational feasibility and proven performance. The non-sparse algorithm's $O((KT)^3)$ complexity is prohibitive, whereas the sparse version's $O(s^2KT^2)$ is practical. More importantly, our results from Simulations II and III already establish that the sparse version performs nearly identically to the exact GP posterior. Given that the sparse algorithm already achieves near-optimal performance in Simulation I (as seen in Figure 2), running the computationally expensive non-sparse version would provide little additional insight.
>
> **Comment:** you should give some reference for the TIM+ oracle.
>
> **Response:** We had cited the TIM+ oracle in Section 5.1.1, from Tang et al., 2014.
>
> **Comment:** "Note that the reason ..." I think this comment was left from...
>
> **Response:** We thank the reviewer for spotting this error that we have fixed in the revised manuscript.
>
> **Comment:** The parameter setting in the second simulation for ...
>
> **Response:** We have revised Section 5.2.2 to be less repetitive and more concise.

---

> ### Author Response · Authors · 2025-07-11
> **Rebuttal**
>
> **Comment:** Figure 6: why not to compare the algorithms over the same environment...
>
> **Response:** We thank the reviewer for this important question, which allows us to clarify the specific goal of Simulation III and Figure 6.
>
> The primary purpose of this experiment is not to provide a direct head-to-head comparison of algorithms in a single, fixed environment. Instead, its goal is to demonstrate how each algorithm's performance scales with increasing inter-arm codependency (controlled by the kernel lengthscale $l$). The current figure design, with one subplot per algorithm, is deliberately chosen because it makes this crucial comparison clear. The key insight is gained by observing the trend within each subplot.
>
> For instance, in the subplots for O'CLOK-UCB and its sparse variants, one can clearly see the performance curves separate and rise significantly as $l$ increases. This visually confirms that our GP-based methods successfully exploit arm codependency to learn more efficiently. In contrast, the subplots for ACC-UCB and CC-MAB show that performance stagnates and does not improve substantially with $l$, demonstrating their inability to leverage this structural information.
>
> Finally, from a practical standpoint, presenting all 30 curves (6 algorithms × 5 $l$ values) on a single graph would create extreme visual clutter, making any meaningful comparison impossible.

---

> ### Comment · Reviewer_7ZVz · 2025-07-21
> **Further Comments**
>
> There are still some (minor) issues that in my opinion should be addressed.
>
> 1) \forall M \leq |S| -> \forall m \in {1, \ldots |S|}
>
> 2) I would add the rationale of the scaled logistic (sigmoid) function in the paper text.
>
> 3) Given your comment on the simulations, I would change the order of the experiments, since some conclusions drawn for Sim. II and III are useful to justify the setting in Sim. I.
>
> 4) You should clarify more the purpose of the experiments in section 5.4. It is stated in the title, but I would add a more explicit statement to explain the goal of the simulation.

---

> > ### Author Response · Authors · 2025-07-23
> > **Response to Further Comments**
> >
> > We thank the reviewer for the further detailed feedback. We have incorporated all suggested changes into the current version of the manuscript. In particular:
> >
> > 1. We have fixed the typo.
> > 2. We have added the rationale for using the sigmoid function in the main paper. We reproduce the added explanation here for convenience:
> >
> > "... to model a realistic non-linear relationship. Its S-shape captures a natural saturation effect, where the outcome is
> > most sensitive to mid-range context values but has diminishing returns at the extremes."
> >
> > 3. Based on the reviewer's suggestion, we have changed the order of the simulations.
> > 4. We have also added a paragraph at the beginning of the section, clarifying the purpose of the experiments there, as below:
> >
> > "The goal of this simulation is to evaluate the performance of our approach against baseline methods under
> > varying levels of base arm codependency, demonstrating how our GP-based method effectively exploits base
> > arm dependencies."

---

### Decision · Action_Editor_b5KB · 2025-08-28

**Recommendation:** Accept as is

**Audience:**

Yes

**Audience Explanation:**

Overall, there certainly is a (limited) audience for this work. In more detail:
- Reviewer 7ZVz believes the claims of the paper are interesting and would be of interest to the combinatorial bandit community. While the algorithm (Algorithm 1) appeared in previous works, the guarantees did not. The authors adequately addressed the reviewers requested changes (although I would prefer the authors use stronger language when discussing the extent to which Algorithm 1 matches previous works: the algorithm seems to be exactly the algorithm presented by Accabi et al. (2018), but the authors’ discussion of this point is very guarded in admitting this.
- Reviewer WMHd mentions that Lemma 1 (which is crucial) is novel, although the overall contribution is incremental. Still, for the limited community operating in this area, there should be sufficient interest in the authors’ results.
- Reviewer yA9y believes the paper has an audience in TMLR and that the theoretical derivations are correct. They mention that Lemma 1 is where the theoretical novelty is, but they are less enthusiastic about the significance of this lemma as compared to previous work.

**Claims And Evidence:**

Yes

**Claims Explanation:**

No reviewers contest the proofs of the theoretical results, and at least one reviewer looked at the proofs in detail. The experimental results also are clear.